# Evident PM$_{2.5}$ Drops in the East of China due to the COVID-19 Quarantines in February

Zhicong Yin [123], Yijia Zhang[1], Huijun Wang[123], Yuyan Li[1]

[1]Key Laboratory of Meteorological Disaster, Ministry of Education / Joint International Research Laboratory of Climate and Environment Change (ILCEC) / Collaborative Innovation Center on Forecast and Evaluation of Meteorological Disasters (CIC-FEMD), Nanjing University of Information Science & Technology, Nanjing, 210044, China

[2]Southern Marine Science and Engineering Guangdong Laboratory (Zhuhai), Zhuhai, 519080, China

[3]Nansen-Zhu International Research Centre, Institute of Atmospheric Physics, Chinese Academy of Sciences, Beijing, China

*Correspondence to*: Zhicong Yin (yinzhc@163.com)

**Abstract.** The top-level emergency response to the COVID-19 pandemic involved an exhaustive quarantine in China. The impacts of COVID-19 quarantine on the decline in fine particulate matter (PM$_{2.5}$) were quantitatively assessed based on numerical simulations and observations in February. Relative to both of February 2017 and climate mean, anomalous southerlies and moister air occurred in the east of China in February 2020, which caused considerable PM$_{2.5}$ anomalies. Thus, it is a must to disentangle the contributions of stable meteorology from the effects of the COVID-19 lockdown. The contributions of routine emission reductions were also quantitatively extrapolated. The top-level emergency response substantially alleviated the level of haze pollution in the east of China. Although climate variability elevated the PM$_{2.5}$ by 29% (relative to 2020 observations), 59% decline related to COVID-19 pandemic and 20% decline from the expected pollution regulation dramatically exceeded the former in North China. The COVID-19 quarantine measures decreased the PM$_{2.5}$ in Yangtze River Delta by 72%. In Hubei Province where most pneumonia cases were confirmed, the impact of total emission reduction (72%) evidently exceeded the rising percentage of PM$_{2.5}$ driven by meteorology (13%).

**Keywords:** COVID-19, PM$_{2.5}$, Emission Reduction, Climate Variability, Haze

## 1 Introduction

The COVID-19 pandemic devastatingly swept through China in the beginning of 2020 (Luo, 2020; Xia et al., 2020; Cao et al., 2020). By April 2020, more than 84 thousand confirmed cases were reported by the National Health Commission of China, approximately 75% of which were confirmed in February (Fig. 1a). To effectively control the large spread of COVID-19 pneumonia, stringent quarantine measures were implemented by the Chinese government and people themselves, including prohibiting social activities, shuttering industries, stopping transportation, etc. (Chen S. et al., 2020). The abovementioned emergency response measures were first carried out in Wuhan on 23 January, which resulted in the delayed arrival of COVID-19 in other cities by 2.91 days, and these response measures were in effect in all cities across China, thus limiting the spread of the COVID-19 epidemic in China (Tian et al., 2020). Since March 7, the number of newly confirmed cases in China has

been nearly below 100. On the other hand, the COVID-19 quarantine measures greatly reduced anthropogenic emissions, and therefore, the air quality in China was considerably improved (Wang et al., 2020). Chen K. et al. (2020) simply compared observations of atmospheric components before and during the quarantine and found that the concentration of fine particulate matter ($PM_{2.5}$) in Wuhan decreased 1.4 μg/m³, but it decreased 18.9 μg/m³ in 367 cities across China. Shi et al. (2020) quantified a 35% reduction of $PM_{2.5}$ on average during the COVID-19 outbreak compared to the pre-COVID-19 period. Huang et al. (2020) used comprehensive measurements and modeling to show that the haze during COVID-19 lockdown was driven by enhancements of secondary pollution, which offset reduction of primary emissions during this period in China. However, the impacts of meteorology on the air quality were neglected in many previous studies.

Climate variability notably influences the formation and intensity of haze pollution in China (Yin and Wang 2016; Xiao et al., 2015; Zou et al., 2017), and the impacts are embodied by variations in surface wind, boundary layer height and moisture conditions (Shi et al., 2019; Niu et al., 2010; Ding et al., 2014). During December 16th-21st 2016, although most aggressive control measures for anthropogenic emissions were implemented, severe haze pollution with $PM_{2.5}$ concentrations ≈ 1100μg m$^{-3}$ still occurred and covered 710,000km². The continuous low surface wind speed of less than 2ms$^{-1}$, high humidity above 80% and strong temperature inversion lasting for 132h caused the serious haze event in 2016 (Yin and Wang, 2017). In winter 2017, the air quality in North China largely improved; however, the stagnant atmosphere in 2018 resulted in a major $PM_{2.5}$ rebound comparing to 2017 by weakening transport dispersion and enhancing the chemical production of secondary aerosols (Yin and Zhang 2020). Wang et al. (2020) applied the Community Multiscale Air Quality model to emphasize that the role of adverse meteorological conditions cannot be neglected even during the COVID-19 outbreak. From February 8 to 13 2020, North China suffered severe pollutions, with maximum daily $PM_{2.5}$ exceeding 200μg m$^{-3}$. During this period, weak southerly surface winds lasted for nearly 5 days, relative humidity was close to 100%, and atmospheric inversion reached more than 10°C. Although pollution emissions from basic social activities have been reduced, heavy pollution still occurred when adverse meteorological conditions characterized by stable air masses appeared (Wang et al., 2020).

After the severe haze events of 2013, routine emission reductions resulted in an approximately 42% decrease in the annual mean $PM_{2.5}$ concentration between 2013 and 2018 in China (Cleaner air for China, 2019). In November 2019, the Ministry of Environmental Protection of China issued a series of Autumn-Winter Air Pollution Prevention and Management Plans indicating that the routine emission reductions would be conventionally implemented in the following winter (Ministry of Environmental Protection of China, 2019). As reported by the government, the mean ratio of work resumption in large industrial enterprises was approximately 90% in the east of China until the end of February (Fig. 1b). In this study, we attempted to quantify the impacts of the COVID-19 pandemic on the observed $PM_{2.5}$ concentration in February 2020 when the quarantine measures were the strictest. The official 7-day Chinese New Year holiday occurs in January and February and commonly accounts for approximately 25% of a month. From 2013–2020, there were only two years (2017 and 2020) when the official 7-day holiday occurred in January (Fig. 1c). Thus, to avoid the impacts of the Spring Festival, the observed $PM_{2.5}$ concentration

in February 2017 (Fig. 1a) was adopted to calculate the PM$_{2.5}$ difference, which was decomposed into the results due to
expected routine emission reductions, changing meteorology climate variability, and COVID-19 quarantines.
**2 Datasets and methods**
**2.1 Data description**

67        Monthly mean meteorological data from 2015 to 2020 were obtained from NCEP/NCAR reanalysis datasets, with a

horizontal resolution of 2.5°×2.5°, including the geopotential height at 500 hPa (H500), zonal and meridional winds at 850
hPa, vertical wind from the surface to 150 hPa, and relative humidity at the surface (Kalnay et al., 1996). PM$_{2.5}$ concentration
data from 2015 to 2020 were acquired from the China National Environmental Monitoring Centre (https://quotsoft.net/air/ ).
The monitoring network expanded from 1500 sites in 2015 to 1640 sites in 2020, covering approximately 370 cities nationwide.
The PM$_{2.5}$ data were monitored every 5 min using two methods: a tapered element oscillating microbalance and β-rays, which
were operated under the China National Quality Control.
**2.2 GEOS-Chem description, evaluation and experimental design.**

75        We used the GEOS-Chem model (http://acmg.seas.harvard.edu/geos/) to simulate the PM$_{2.5}$ concentration, driven by

MERRA-2 assimilated meteorological data (Gelaro et al., 2017). The nested grid over China (15° N–55° N, 75–135° E) had a
horizontal resolution of 0.5° latitude by 0.625° longitude and consisted of 47 vertical layers up to 0.01 hPa. The GEOS-Chem
model included the fully coupled O$_3$–NO$x$–hydrocarbon and aerosol chemistry module with more than 80 species and 300
reactions (Bey et al., 2001; Park et al., 2004). The PM$_{2.5}$ components simulated in the GEOS-Chem model included sulfate,
nitrate, ammonium, black carbon and primary organic carbon, mineral dust, and sea salt. Aerosol thermodynamic equilibrium
is computed by the ISORROPIA package, which calculates the gas–aerosol partitioning of the sulfate–nitrate–ammonium
system (Fountoukis and Nenes, 2007). Heterogeneous reactions of aerosols include the uptake of HO$_2$ by aerosols (Thornton
et al., 2008), irreversible absorption of NO$_2$ and NO$_3$ on wet aerosols (Jacob, 2000), and hydrolysis of N$_2$O$_5$ (Evans and Jacob,
2005). Two alternate simulations of aerosol microphysics are implemented in GEOS-Chem: the TOMAS simulation (Kodros
and Pierce, 2017) and the APM simulation (Yu and Luo, 2009), which were both simulated in the experiments.

86        GEOS-Chem model has been widely used to examine the historical changes in air quality in China and quantitatively

separate the impacts of physical-chemical processes. Using the GEOS-Chem model, Yang et al. (2016) found an increasing
trend of winter PM$_{2.5}$ concentrations during 1985–2005, 80% of which due to anthropogenic emissions and 20% due to
meteorological conditions. Here, we simulated the PM$_{2.5}$ concentrations in February 2017 and evaluated the performance of
GEOS-Chem (Fig. 2a). The values of mean square error / mean equals were 5.8%, 7.0% and 5.4% in North China (NC),
Yangtze River Delta (YRD) and Hubei Province (HB), respectively, indicating the good performance of reproducing the haze-
polluted conditions. The absolute biases were larger in the south of China, which was consistent with Dang and Liao (2019).
They also compared the simulated and observed daily mean $PM_{2.5}$ concentrations at the Beijing, Shanghai, and Chengdu grids,
which had a low bias in Beijing with a normalized mean bias (NMB) of -9.2% and high biases with NMBs of 18.6% and 28.7%
in Shanghai and Chengdu, respectively. The simulations in February 2017 in this study substantially underestimated the $PM_{2.5}$
in NC with an NMB of -3.0% (Fig. 2a). Among them, the NMB in The Beijing-Tianjin-Hebei region was -3.3%. However, in
the Fenwei plain, the underestimation was even more pronounced, with NMB reaching -16.3%. The simulated biases possibly
affected the subsequent results and brought uncertainties to some extent. The simulated spatial distribution of $PM_{2.5}$ was also
similar to that of observations with spatial correlation coefficient = 0.78.
We further verified whether the simulations could capture the roles of meteorological changes in February 2020 under a
substantial reduction in emissions because of COVID-19 quarantines. In NC, YRD and HB, the correlation coefficients
between daily $PM_{2.5}$ observations and simulated data under 2010 (1985) emission scenario reached 0.83 (0.82), 0.67 (0.63),
and 0.79 (0.73), respectively (Fig. 2b-d), and could capture the maximum and minimum $PM_{2.5}$ concentrations. For example,
in NC, the simulation could well simulate severe haze events (e.g., from 8–13 and 19–25 February) and good air quality events
(e.g., from 14–18 February), reflecting that it has ability to accurately capture the change of meteorological conditions. The
correlation coefficients under 2010 emission scenario were all higher than that under 1985 emission scenario maybe due to the
emissions from each sector in 2010 were more similar to recent years, which was more reasonable.
The $PM_{2.5}$ concentration in February from 2015 to 2020 was simulated in this study. Due to delayed updates of the
emission             inventory,             we             used             the             emissions             data             of             2010
(http://geoschemdata.computecanada.ca/ExtData/HEMCO/AnnualScalar) and 1985 (M. Li et al., 2017) for the simulations,
which represented high- and low-emission scenarios, respectively. In total, we conducted two sets of numerical experiments
to drive the GEOS-Chem simulations, one combining the meteorological conditions from 2015 to 2020 with fixed emissions
in 1985 and the other with fixed emissions in 2010, which could determine the stability of simulated results.
**2.3 The method to quantify the influence of the COVID-19 quarantine.**
As mentioned above, we aimed to examine the impact of the COVID-19 quarantines on $PM_{2.5}$ over the February 2017
level basing on an observational-numerical hybrid method. The observed $PM_{2.5}$ difference in February 2020 ($PMd_{OBS}$) was
linearly decomposed into three parts: the impacts of changing meteorology ($PMd_M$), expected routine emissions reductions
($PMd_R$) and COVID-19 quarantines ($PMd_C$), which was a reasonable approximation, and the decomposition equation was
$PMd_{OBS} = PMd_M + PMd_R + PMd_C$. That is, $PMd_C = PMd_{OBS} - PMd_M - PMd_R$. It should be noted that $PMd_C$ is the impact of
the COVID-19 quarantines over the situation whereby the pandemic did not occur and routine emission reductions
conventionally were in effect. The value of $PMd_E$ (i.e., $PMd_R + PMd_C$) was the total impact of the emission reductions in
February 2020 over the 2017 level.
Simulated $PM_{2.5}$ data driven by changing meteorology with two fixed-emissions (1985 and 2010) were employed to
determine the ratio of $PMd_M$ of each year/ observed $PM_{2.5}$ in 2017. Depending on the GEOS-Chem simulations, we found that
the percentage of changed $PM_{2.5}$ due to the differences in meteorology remained nearly constant regardless of the emission
level (Fig. S1), which was consistent with the results of Yin and Zhang (2020). This percentage was the difference of simulated
$PM_{2.5}$ between each year and 2017 under the same emission scenario divided by the simulated $PM_{2.5}$ in 2017. For example, the
percentages due to different meteorology between 2020 and 2017 were 22.1% (21.4%), −1.2% (−0.7%) and 9.0% (8.2%) in
NC, YRD and HB under the low (high) emissions (Fig. S1). The percentage under 2010 emission scenario was selected as the
final percentage because the emissions from each sector in 2010 were more similar to recent years, and thus was more
reasonable. Then, through multiplying the 2017 observation by this percentage, $PMd_M$ can be quantified in each simulation
grid with respect to 2017 (STEP 1).
From 2015 to 2019, $PMd_C = 0$; thus, $PMd_R = PMd_{OBS} - PMd_M$. Here, we repeated STEP 1 to determine $PMd_M$ in each year
from 2015 to 2019 relative to 2017 (i.e., $PMd_M = 0$ in 2017). After removing the effect of meteorological conditions in $PM_{2.5}$
differences, $PMd_R$ in all years except 2020 can also be calculated. According to many previous studies, the change in emissions
resulted in a linear change in air pollution in China from 2013-2019 (Wang et al., 2020; Geng et al., 2020) which might be
related to the huge emission reduction due to the implementation of clean air action. Because the signal of emissions reduction
in China had been particularly strong since 2013, it could be easily detected and the assumption of a linear reduction in
pollution caused by emission reduction was applicable in China in the past few years. Based on this approximation, we used
the method of extrapolation to speculate the impact of routine emission reduction on $PM_{2.5}$. We performed linear extrapolation
based on known $PMd_R$ values from 2015 to 2019 to obtain $PMd_R$ in 2020 (STEP 2, Fig. S2). This $PMd_R$ in 2020was calculated
as the change of $PM_{2.5}$ caused by expected routine emission reduction, which did not actually happen, but merely gave an
assessment in the case of "if no COVID-19". In Beijing and Shanghai, for example, $PM_{2.5}$ fell by 23.1% and 26.6% due to
routine emission reduction in 2019, respectively, compared with 2015. Zhou et al. (2020) indicated that emission reductions
caused 20–26% decreases in winter in Beijing which has been translated into 5 years. Zhang et al. (2020) also showed that the
emission controls in Beijing-Tianjin-Hebei (BTH) region have led to significant reductions in $PM_{2.5}$ from 2013 to 2017 of
approximately 20 % after excluding the impacts of meteorology. Geng et al. (2020) found a 20% drop in the main component
of $PM_{2.5}$ in the Yangtze River Delta from 2013 to 2017. These results are consistent with our extrapolated results. Therefore,
it is reasonable to obtain $PMd_R$ by extrapolation after disentangling the effects of meteorological conditions.
Through STEP 1 and STEP 2, $PMd_C$ and $PMd_R$, respectively, in 2020 can be determined. $PMd_{OBS}$ can be directly
calculated from the observed data. After removing the influences of climate anomalies and routine emission reductions, the
impact of COVID-19 quarantines on $PM_{2.5}$ ($PMd_C$) was extracted as $PMd_{OBS} - PMd_M - PMd_R$ (STEP 3).

## 3 Results

The mean $PM_{2.5}$ concentration in February 2020 was nearly below 80 μg/m³ at the vast majority of sites in the east of China, which was much lower than before (Fig. S3). North China (NC) was still the most polluted region (>40 μg/m³), but the $PM_{2.5}$ concentrations in the Pearl River Delta (PRD) and Yangtze River Delta (YRD) were < 20 μg/m³ and < 40 μg/m³, respectively. Relative to the observations in February 2017, negative $PM_{2.5}$ anomalies were centered in NC, with values of approximately –60 to –40 μg/m³ in southern Hebei Province and northern Henan Province (Fig. 3). In Hubei Province (HB), where the COVID-19 pneumonia cases were the most severe in February, the $PM_{2.5}$ concentration was 20~40 μg/m³ lower than that in 2017. The $PM_{2.5}$ differences were also negative in YRD and PRD. Therefore, how much did air pollution decrease due to the COVID-19 quarantines in February in east of China?

Climate variability notably influences the interannual-decadal variations in haze pollution as verified by both observational analysis (Yin et al., 2015) and GEOS-Chem simulations (Dang and Liao, 2019). Furthermore, Zhang et al. (2020) reported that meteorology contributes 50% and 78% of the wintertime $PM_{2.5}$ reduction between 2017 and 2013 in the BTH and YRD, respectively. Therefore, it is necessary to disentangle the influences of climate anomalies before quantifying the contributions of the COVID-19 quarantines on the air quality. The highest observed $PM_{2.5}$ concentrations were 274, 223, and 303 μg/m³ in Beijing, Tianjin and Shijiazhuang, respectively. Although human activities had sharply decreased, severe haze pollution (e.g., 8–13 and 19–25 February 2020) was not avoided, which was attributed to the stagnant atmosphere (Wang et al., 2020), and these severe haze events were also reproduced by the GEOS-Chem simulation (see Section 2.2 and Fig. 2b).

As shown in Figure 4a-b, the meteorological conditions in February 2020 were more favorable for the occurrence of haze pollution in NC. In the mid-troposphere, an anomalous anticyclone was located over NC and the Sea of Japan (Fig. 4a). These anticyclonic anomalies clearly stimulated anomalous southerlies over eastern China, which not only transported sufficient water vapor to NC but also overwhelmed the climatic northerlies in winter (Fig. 4b). In addition, the anomalous upward motion associated with anomalous anticyclones prevented the downward transportation of westerly momentum and preserved the thermal inversion layer over NC (Fig. S4). Particularly, in the stagnant days (i.e., 8–13 and 19–25 February), the East Asia deep trough, one of the most significant zonally asymmetric circulations in the wintertime Northern Hemisphere (Song et al., 2016), shifted eastwards and northwards than climate mean, which steered the cold air to North Pacific instead of North China (Fig. 4c). The climatic northerlies in February, related to East Asia winter monsoon, also turned to be south winds in the east of China (Fig. 4d). Physically, the weakening surface winds and strong thermal inversion corresponded to weaker dispersion conditions, and the higher humidity indicated a favorable environment for the hygroscopic growth of aerosol particles to evidently decrease the visibility. Compared with the climate (February 2017) monthly mean, boundary layer height (BLH) decreased by 19.5m (34.5m), surface relative humidity (rhum) increased by 5% (10.6%) and surface air temperature (SAT) rose by 1.6°C (0.9°C) after detrending, which were conductive to the increase of $PM_{2.5}$ concentration in February 2020.

Furthermore, the correlation coefficients of daily $PM_{2.5}$ and BLH, rhum, wind speed and SAT in North China were -0.63, 0.44,
-0.45 and 0.46, respectively, all of which passed the 95% significance test using $t$ test method and indicated importance of
meteorology. We used the meteorological data in February 2017 to establish a multiple linear regression equation to fit $PM_{2.5}$.
The correlation coefficients between the fitting results and the observed $PM_{2.5}$ concentration in NC, YRD and HB reached 0.84,
0.64 and 0.65, exceeding the 99% significance test using $t$ test method. Then, we put the observed meteorological data in
February 2020 into this established multiple regression equation to get the predicted $PM_{2.5}$ concentration. Using the regress-
predicted value, the percentage of changed $PM_{2.5}$ due to the differences in meteorology between 2017 and 2020 were re-
calculated and is 20.7%, -3.2% and 9.5% in NC, YRD and HB, respectively (Fig. S1), which is consistent with and enhanced
the robustness of the results obtained by our previous model simulation. Based on the GEOS-Chem simulations, $PMd_M$ was
calculated between February 2020 and 2017 (see Methods). To the south of 30°N, most $PMd_M$ values were negative with small
absolute values, at < 10 μg/m$^3$. To the north of 30°N, the $PMd_M$ values were mostly positive, ranging from 30~60 μg/m$^3$ in
BTH (Fig. 5a).

196        Since 2013, the Chinese government has legislated and implemented stringent air pollution prevention and management

policies that have clearly contributed to air quality improvement (Wang et al., 2019). As mentioned above, without the COVID-
19 pandemic, these emission reduction policies would certainly remain in effect in February 2020. Thus, we extrapolated $PMd_R$
(i.e., the $PM_{2.5}$ difference due to expected routine emission reductions) between February 2020 and 2017 to isolate the impacts
of the COVID-19 quarantines (i.e., $PMd_C$). $PMd_R$ was mostly negative in the east of China (Fig. 5b). Because the impacts of
meteorology were proactively removed, these negative values illustrated that routine emission reductions substantially reduced
the wintertime $PM_{2.5}$ concentration. The contributions of the emission reduction policies were the greatest in the south of BTH
and were also remarkable in Hubei Province (Fig. 5b). Although the $PMd_R$ of Beijing in 2016 did not strictly comply with the
pattern of monotonous decrease, which might be caused by the fluctuation of policy and its implementation, the value of $PMd_R$
in 2020 relative to 2017 was –8.4 μg/m$^3$ and was comparable to the 11.5 μg/m$^3$ reductions due to policy during 2013–2017
(Zhang et al., 2020). In Shanghai, $PMd_R$ was –12.0 μg/m$^3$ (Fig. 6), whose magnitude was proportional with assessments by
Zhang et al. (2020), and the trend was nearly linear. The rationality of the extrapolations of $PMd_R$ was also proved in Section
2.3. The trend of $PMd_R$ in Wuhan was –9.6 μg/m$^3$ per year from 2015–2019, which indicated high efficiency of the emission
reduction policies and resulted in large $PMd_R$ values in 2020 (i.e., –21.8 μg/m$^3$).

210        By disentangling the impacts of meteorology and routine emission reduction policies, the change in $PM_{2.5}$ due to the

COVID-19 quarantines was quantitatively extracted. As expected, this severe pandemic caused dramatic slumps in the $PM_{2.5}$
concentration across China (Fig. 5c). Large $PMd_C$ values (approximately –60 to –30 μg/m$^3$) were located in the high-polluted
NC regions where intensive heavy industries were stopped and the traditional massive social activities and transportations
around Chinese New Year were cancelled as part of the COVID-19 quarantine measures. To the south of 30°N, the impacts of
the COVID-19 quarantines on the air quality were relatively weaker (–30 ~ 0 μg/m$^3$) than those in the north. Generally, the

south region was less polluted than the north, therefore the baseline of $PM_{2.5}$ concentration was relatively lower (Fig. S3a). In addition, meteorological conditions in the south in February 2020 had no positive contribution (Fig. 5a), which would not lead to the increase of $PM_{2.5}$ concentration. These two possible reasons resulted in a smaller space for $PM_{2.5}$ decrease due to COVID-19 quarantines in the south and accompanying regional differences. To reduce the assessment uncertainties, the percentage of changed $PM_{2.5}$ due to the differences in meteorology were recalculated based on the GEOS-Chem simulations with fixed emission in 1985. As described in the Methods section, the recalculated $PMd_C$ in Figure S5 were consistent with those in Figure 5c, showing a high robustness. Furthermore, the mean $PM_{2.5}$ concentration decreases due to the COVID-19 quarantines in NC, HB and YRD were analyzed, which accounted for 59%, 26% and 72% of the observed February $PM_{2.5}$ concentration in 2020 (Fig. 7).

It should be noted that the sum of $PMd_R$ and $PMd_C$ (i.e., $PMd_E$) is the total contribution of the emission reduction in February 2020 with respect to 2017 (Fig. 5d). In NC, YRD and HB, the COVID-19 quarantines and routine emission reductions drove $PM_{2.5}$ in the same direction. The mean $PM_{2.5}$ decrease in NC, due to the total emission reduction, was $-43.3$ μg/m$^3$, accounting for 79% of the observed February $PM_{2.5}$ concentration in 2020 (Fig. 7). Although the absolute values of both $PMd_R$ and $PMd_C$ in YRD were smaller than those in NC, the change percentage (92%) was larger because of the lower base $PM_{2.5}$ concentration. In HB, where more than 80% of the confirmed COVID-19 cases in China occurred and the cities were in emergency lockdown, the total anthropogenic emissions were clearly limited, which resulted in a 72% decline in $PM_{2.5}$ in the atmosphere (Fig. 7). In particular, if the anthropogenic emissions did not decline, the $PM_{2.5}$ concentration in NC, YRD and HB would increase to nearly twice the current observation (Fig. 7), indicating significant contributions of human activities to the air pollution in China.

The declines of $PM_{2.5}$ seemed not to be directly proportional to the almost complete shutoff of vehicle traffics and industries, that is, the reduction ratio of $PM_{2.5}$ concentrations were smaller than that of precursor emissions (Wang et al., 2020). The unexpected air pollutions during the marked emission reductions were closely related to the stagnant air flow, enhanced productions of secondary aerosols, and uninterrupted residential heating, power plants and petrochemical facilities (Le et al., 2020). The partial impacts of stagnant meteorological conditions have been explained earlier (Fig. 4). In Wuhan, the $PM_{2.5}$ remained the main pollutant during the city lockdown and the high level of sulphur dioxide ($SO_2$) may be related to the increased domestic heating and cooking (Lian et al., 2020). In North China, large reductions of primary aerosols were observed, but the decreases in secondary aerosols were much smaller (Sun et al., 2020; Shi et al., 2020). Because of the disruption of transportations, reduced nitrogen oxide ($NOx$) increased the concentrations of ozone and nighttime nitrate ($NO_3$) radical formations. The increased oxidizing capacity in the atmosphere enhanced the formation of secondary particulate matters (Huang et al., 2020). Thus, the non-linear relationship of emission reduction and secondary aerosols also partially contributed to the haze occurrence during the COVID-19 lockdown.

**4 Conclusions and discussion**

In the beginning of 2020, the Chinese government implemented top-level emergency response measures to contain the spread of COVID-19. The traditional social activities surrounding Chinese New Year, industrial and transportation activities, etc. were prohibited, which effectively reduced the number of confirmed cases in China. Concomitantly, anthropogenic emissions, which are the fundamental reason for haze pollution, were dramatically reduced by the COVID-19 quarantine measures. In this study, we employed observations and GEOS-Chem simulations to quantify the impacts of the COVID-19 quarantines on the air quality improvement in February 2020 after decomposing the contributions of expected routine emission reductions and climate variability. Although the specific influences varied by the region, the COVID-19 quarantines substantially decreased the level of haze pollution in the east of China (Fig. 7). In North China, the meteorological conditions were stagnant that enhanced the $PM_{2.5}$ concentration by 30% (relative to the observations in 2020). In contrast, the expected routine emissions reductions and emergency COVID-19 quarantine measures resulted in an 80% decline. In YRD, the impacts of meteorology were negligible but the COVID-19 quarantines decreased $PM_{2.5}$ by 72%. In Hubei Province, the impact of the total emission reduction (72%) evidently exceeded the $PM_{2.5}$ increase due to meteorological conditions (13%). In March, due to the continued control of the COVID-19, the quarantines measures still contributed to the negative anomalies of the observed $PM_{2.5}$ between 2020 and 2017 (Fig. 8a). Because the activities in production and life have been gradually resumed in March, the $PM_{2.5}$ drops caused by the COVID-19 quarantines became weaker compared with February (Fig. 8b, c). The contributions of $PMd_C$ to the change of $PM_{2.5}$ concentration in NC, YRD and HB declined from 32.2, 21.0 and 12.1 μg/m$^3$ in February to 7.0, 2.4 and 6.7 μg/m$^3$ in March respectively.

Because of the common update delay of the emission inventory, we employed a combined analysis consisting of observational and numerical methods. We strictly demonstrated the rationality of this method and the results, mainly based on the relatively constant contribution ratio of changing meteorology from GEOS-Chem simulations under the different emissions (Yin and Zhang 2020). However, there was a certain bias in the simulations by GEOS-Chem model, and the biases also showed regional differences (Dang and Liao, 2019). Therefore, gaps between the assessed results and reality still exist, which requires further numerical experiments when the emission inventory is updated. Furthermore, during the calculation process, the observed $PM_{2.5}$ difference in February 2020 was linearly decomposed into three parts. Although this linear decomposition was reasonable in China in the past few years, we must note that this approximation did not consider the meteorology-emission interactions, the product of the emission, the loss lifetime and particularly the sulfate-nitrate-ammonia thermodynamics (Cai et al., 2017), which brought some uncertainties. The actual emission reduction effect is considerable (Fig. 3d), in line with the increasingly strengthened emission reduction policies in recent years. When calculating the $PMd_R$ in 2020, we use the method of extrapolation. Although the result is consistent with others observational and numerical studies (Geng et al., 2020; Zhang et al., 2020; Zhou et al., 2019), it is still estimated value rather than true value. These issues need to be examined in the future

studies to unlock respective effects of emissions and meteorological conditions on $PM_{2.5}$ over eastern China. To restrict the
possible uncertainties, we set up some constraints: 1. The pivotal contribution ratio of changing meteorology were calculated
under two emission levels and recalculated by statistical regressed model; 2. The values of $PMd_M$ and $PMd_R$ were widely
compared to previous studies.
If the COVID-19 epidemic did not occurred, the concentrations of $PM_{2.5}$ would increase up to 1.3–1.7 times the
observations in February 2020 (Fig. 7). Therefore, the pollution abatement must continue. Because of the huge population base
in the east of China, the anthropogenic emissions exceeded the atmospheric environmental capacity even during COVID-19
quarantines. Although the $PM_{2.5}$ dropped much, marked air pollutions also occurred during this unique experiments that the
human emissions were sharply closed. This raised new scientific questions, such as changes of atmospheric heterogeneous
reactions and oxidability under extreme emission control, quantitative meteorology-emission interactions, and so on. This also
implied reconsiderations of policy for pollution controls and necessity to cut off secondary productions of particulate matters
basing on sufficient scientific research (Le et al., 2020; Huang et al., 2020). Some studies estimated that thousands of deaths
were prevented during the quarantine because of the air pollution decrease (Chen K. et al., 2020). However, medical systems
were still overstressed, and transportation to hospitals also decreased. Furthermore, the deaths related to air pollution were
almost all due to respiratory diseases (Wang et al., 2001), and their corresponding medical resources were also further stressed
by COVID-19. Therefore, the mortality impacted by the air pollution reduction during the COVID-19 outbreak should be
comprehensively assessed in future work.
***Data availability.*** Monthly mean meteorological data are obtained from ERA5 reanalysis data archive:
https://cds.climate.copernicus.eu/cdsapp#!/search?type=dataset. $PM_{2.5}$ concentration data are acquired from the China
National Environmental Monitoring Centre: http://beijingair.sinaapp.com/. The emissions data of 1985 can be downloaded
from http://geoschemdata.computecanada.ca/ExtData/HEMCO/AnnualScalar/, and that of 2010 can be obtained from MIX:
http://geoschemdata.computecanada.ca/ExtData/HEMCO/MIX.
**Acknowledgements**
The National Natural Science Foundation of China (41991283, 9174431 and 41705058), the funding of Jiangsu innovation &
entrepreneurship team, and the special project "the impacts of meteorology on large-scale spread of influenza virus" from CIC-
FEMD supported this research.
**Authors' contribution**
Wang H. J. and Yin Z. C. designed and performed researches. Zhang Y. J. simulated the $PM_{2.5}$ by GEOS-Chem model and Li
Y. Y. did the statistical analysis. Yin Z. C. prepared the manuscript with contributions from all co-authors.
**Competing interests**
The authors declare no conflict of interest.

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

**Figure Captions**
Figure 1. (a) Variation in existing confirmed cases (bar; red: increase, blue: decrease) and the ratio of accumulated confirmed
cases to total confirmed cases (black line) in China. (b) The ratio of work resumption in large industrial enterprises in the east
of China. (c) Time of the official 7-days holiday of Chinese New Year from 2013 to 2020.
Figure 2. (a) Spatial distribution of observed (dots) and GEOS-Chem simulated (shading) PM$_{2.5}$ (unit: μg/m$^3$) in February
2017. Observed PM$_{2.5}$ concentrations (black, unit: μg/m$^3$) and simulated PM$_{2.5}$ concentrations under 2010 emission (red) and
1985 emission (blue) in February 2020 in (b) North China (NC), (c)Yangtze River Delta (YRD) and (d) Hubei Province (HB).
Figure 3. Differences in the observed PM$_{2.5}$ (unit: μg/m$^3$) in February between 2020 and 2017. The black boxes indicate the
locations of North China (NC, 32.5-42°N,110-120°E), the Yangtze River Delta (YRD, 28-32.5°N,118-122°E) and Hubei
Province (HB, 30-32.5°N,109.5-116°E).
Figure 4. Differences in the observed atmospheric circulation in February between 2020 and 2017, including (a) geopotential
potential height at 500 hPa (unit: gpm), (b) wind at 850 hPa (arrows; unit: m/s), surface relative humidity (shading; unit: %).
The atmospheric circulations in the stagnant days (e.g., from 8–13 and 19–25 February 2020) were also showed, including (c)

geopotential potential height at 500 hPa (shading; unit: gpm) and its climate mean in February (contour), and (d) wind at 850 hPa (black arrows; unit: m/s), its climate mean (blue arrows) and the increased surface relative humidity (shading; unit: %, stagnant days minus climate mean).

Figure 5. $PM_{2.5}$ difference (unit: μg/m$^3$) in February between 2020 and 2017 due to (a) changing meteorology ($PMd_M$), (b) expected routine emission reductions ($PMd_R$), (c) the COVID-19 quarantines ($PMd_C$), and (d) due to the total emission reduction ($PMd_E = PMd_R + PMd_C$).

Figure 6. Variation in $PMd_R$ (unit: μg/m$^3$) with respect to the February 2017 level in Beijing, Shanghai and Wuhan from 2015 to 2019. $PMd_R$ in 2020 was linearly extrapolated from that in the 2015–2019 period. The dotted line is the linear trend.

Figure 7. Contributions of $PMd_M$ (orange bars with hatching), $PMd_R$ (purple bars with hatching) and $PMd_C$ (blue bars with hatching) to the change in $PM_{2.5}$ concentration (unit: μg/m$^3$) between 2020 and 2017 in the three regions. The observed $PM_{2.5}$ concentration in February 2017 (black) and 2020 (gray) was also plotted, and the expected $PM_{2.5}$ concentration without the COVID-19 quarantine is indicated by black hollow bars. The contribution ratios of the three factors (relative to the $PM_{2.5}$ observations in 2020) are also indicated on the corresponding bars.

Figure 8. (a) Differences in the observed $PM_{2.5}$ (unit: μg/m3) in March between 2020 and 2017. (b) Contributions of $PMd_C$ to the change in $PM_{2.5}$ concentration (unit: μg/m$^3$) between 2020 and 2017 and (c) the contribution ratios of $PMd_C$ (relative to the $PM_{2.5}$ observations in 2020) in March (blue) and February (red) in the three regions.

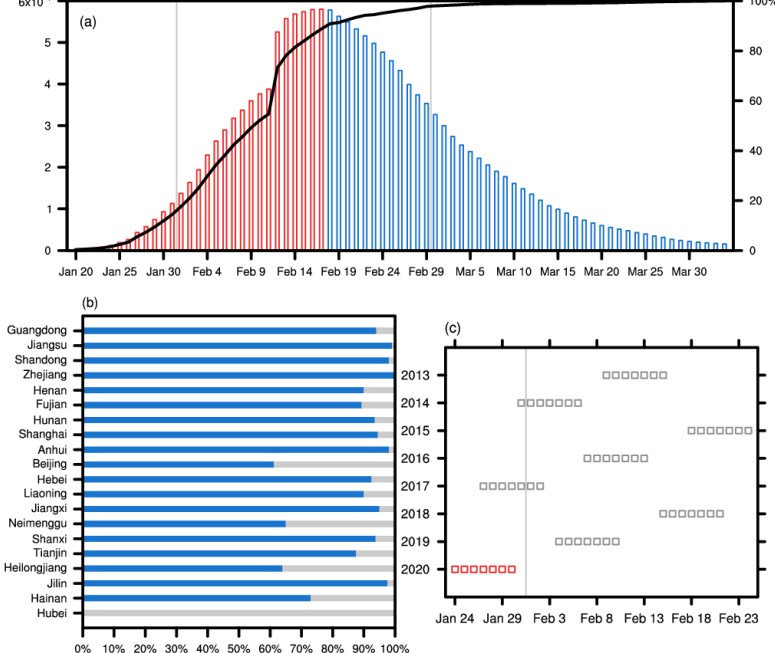


**Figure 1.** (a) Variation in existing confirmed cases (bar; red: increase, blue: decrease) and the ratio of accumulated confirmed cases to total confirmed cases (black line) in China. (b) The ratio of work resumption in large industrial enterprises in the east of China until the end February. (c) Time of the official 7-days holiday of Chinese New Year from 2013 to 2020.

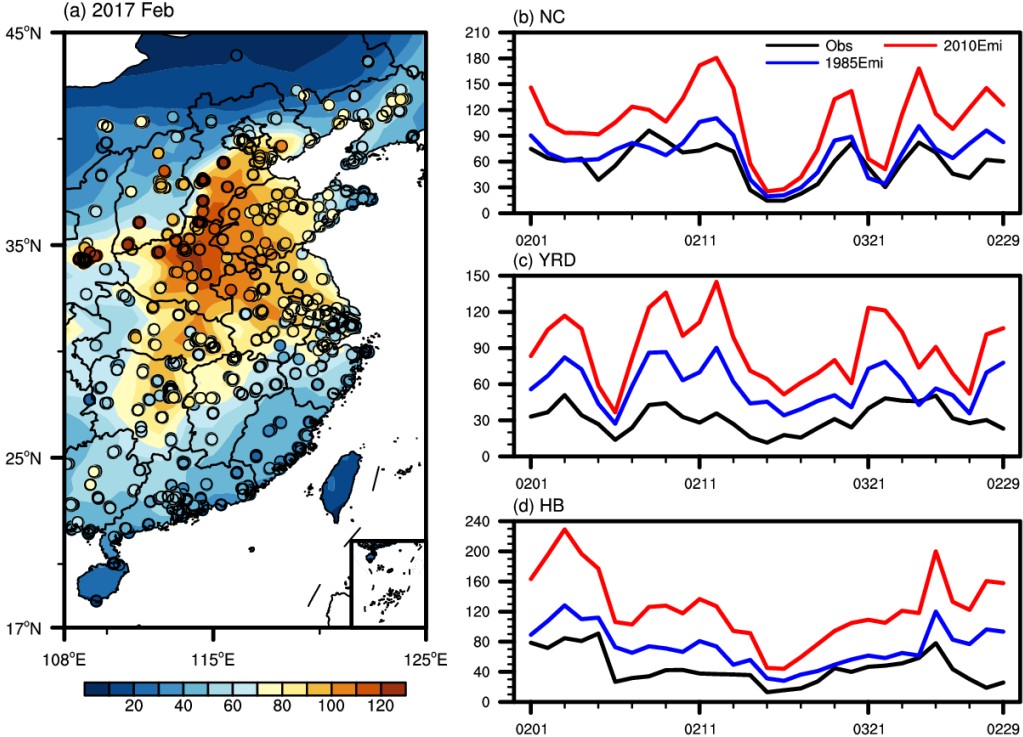


**Figure 2.** (a) Spatial distribution of observed (dots) and GEOS-Chem simulated (shading) PM$_{2.5}$ (unit: μg/m$^3$) in February 2017. Observed PM$_{2.5}$ concentrations (black, unit: μg/m$^3$) and simulated PM$_{2.5}$ concentrations under 2010 emission (red) and 1985 emission (blue) in February 2020 in (b) North China (NC), (c)Yangtze River Delta (YRD) and (d) Hubei Province (HB).

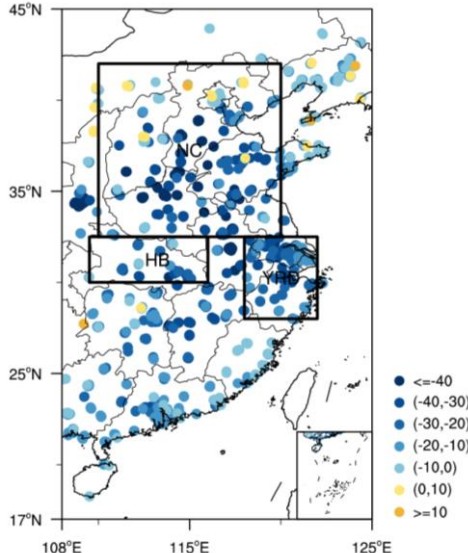


**Figure 3.** Differences in the observed PM$_{2.5}$ (unit: μg/m$^3$) in February between 2020 and 2017. The black boxes indicate the

locations of North China (NC, 32.5-42°N,110-120°E), the Yangtze River Delta (YRD, 28-32.5°N,118-122°E) and Hubei

Province (HB, 30-32.5°N,109.5-116°E).

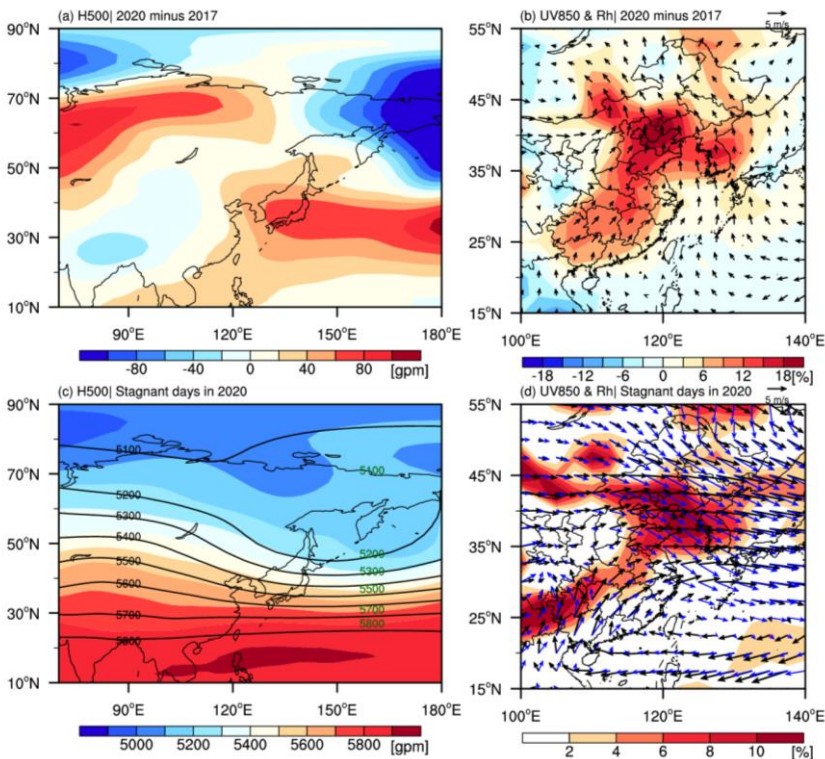


**Figure 4.** Differences in the observed atmospheric circulation in February between 2020 and 2017, including (a) geopotential

potential height at 500 hPa (unit: gpm), (b) wind at 850 hPa (arrows; unit: m/s), surface relative humidity (shading; unit: %).

The atmospheric circulations in the stagnant days (e.g., from 8–13 and 19–25 February 2020) were also showed, including (c)

geopotential potential height at 500 hPa (shading; unit: gpm) and its climate mean in February (contour), and (d) wind at 850

hPa (black arrows; unit: m/s), its climate mean (blue arrows) and the increased surface relative humidity (shading; unit: %,

stagnant days minus climate mean).

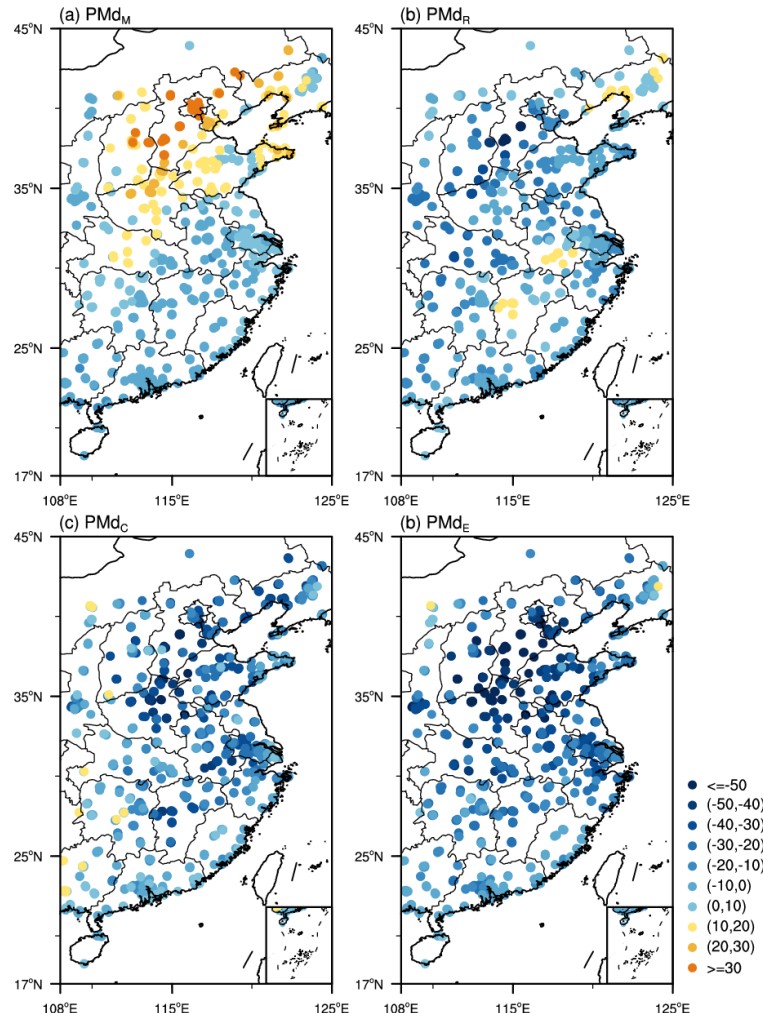

473

**Figure 5.** PM$_{2.5}$ difference (unit: μg/m$^3$) in February between 2020 and 2017 due to (a) changing meteorology (PMd$_M$), (b) expected routine emission reductions (PMd$_R$), (c) the COVID-19 quarantines (PMd$_C$), and (d) due to the total emission reduction (PMd$_E$ = PMd$_R$+ PMd$_C$).

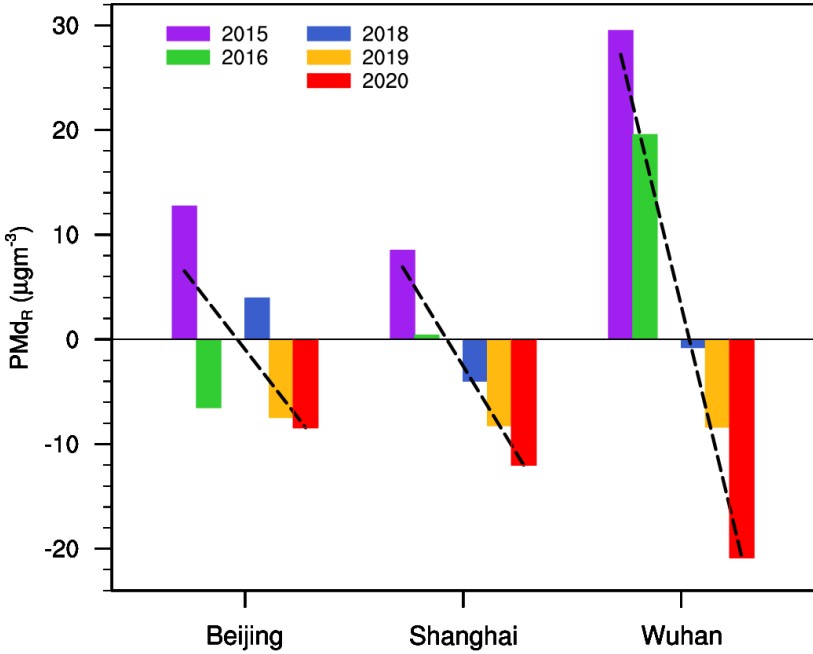


**Figure 6.** Variation in $PMd_R$ (unit: μg/m³) with respect to the February 2017 level in Beijing, Shanghai and Wuhan from 2015

to 2019. $PMd_R$ in 2020 was linearly extrapolated from that in the 2015–2019 period. The dotted line is the linear trend.

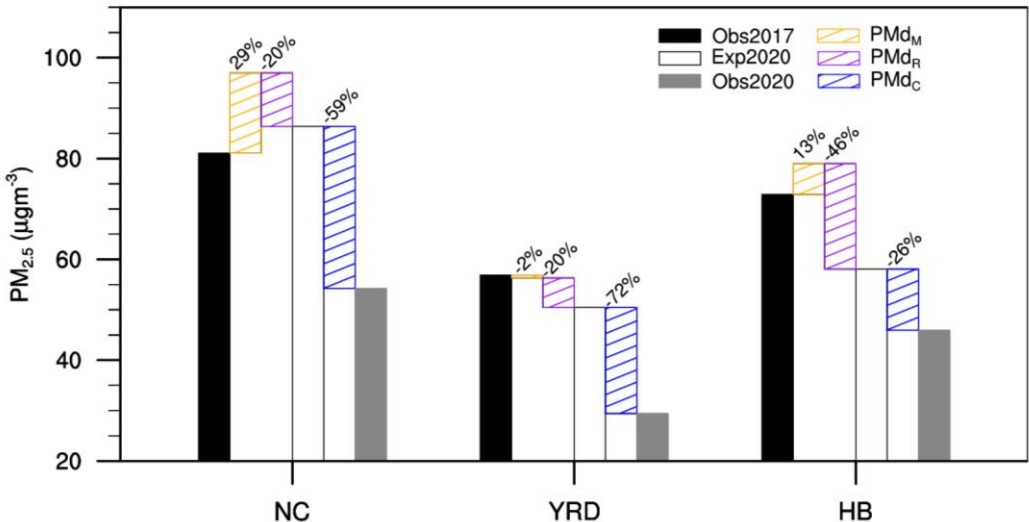


**Figure 7.** Contributions of $PMd_M$ (orange bars with hatching), $PMd_R$ (purple bars with

hatching) and $PMd_C$ (blue bars with hatching) to the change in $PM_{2.5}$ concentration (unit: μg/m³) between 2020 and 2017 in the three regions. The observed $PM_{2.5}$
concentration in February 2017 (black) and 2020 (gray) was also plotted, and the expected $PM_{2.5}$ concentration without the
COVID-19 quarantine is indicated by black hollow bars. The contribution ratios of the three factors (relative to the $PM_{2.5}$
observations in 2020) are also indicated on the corresponding bars.

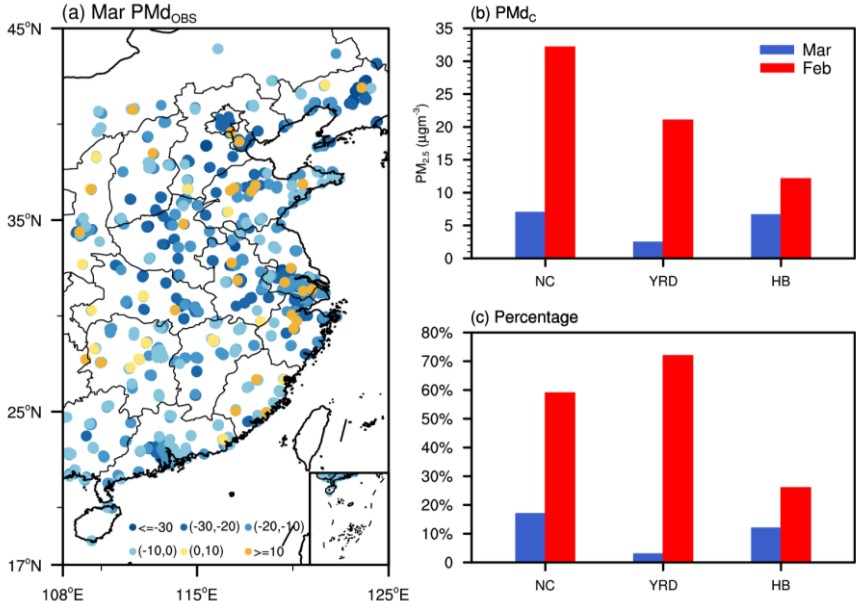

**Figure 8.** (a) Differences in the observed PM$_{2.5}$ (unit: μg/m$^3$) in March between 2020 and 2017. (b) Contributions of PMd$_C$ to the change in PM$_{2.5}$ concentration (unit: μg/m$^3$) between 2020 and 2017 and (c) the contribution ratios of PMd$_C$ (relative to the PM$_{2.5}$ observations in 2020) in March (blue) and February (red) in the three regions.