# Peer review of "Evident PM2.5 Drops in the East of China due to the COVID-19 2 Quarantines in February"

_Atmospheric Chemistry and Physics, 2020_

## Referee Comment (RC1) · Anonymous Referee #1 · 17 Sep 2020

The authors simulate the decline in PM2.5 concentration that resulted from emissions reductions during the COVID-19 pandemic using GEOS-chem. They use 1985 and 2010 emissions to simulate the 2015-20 period. They obtain reasonably good correlations between simulated and observed daily mean PM2.5 and show that COVID-19 led to a significant decline.

The study is interesting, in the sense that knowing how much PM2.5 declined due to COVID-19 after other factors are accounted for is useful, and well-timed. The physical and chemical processes responsible for PM2.5 concentrations during COVID are discussed to some extent.

In response to my comments during the access review, the authors added two new subfigures elucidating the role of meteorology in generating PM2.5, and they added a

literature review of chemical mechanisms for the formation of the remaining pollution. These additions are valuable, but in my opinion further major revisions are still needed before the paper can be published, as follows:

1. Abstract and introduction.

The abstract and introduction should be refocused towards atmospheric processes. While atmospheric processes are discussed (lines 30-37 and 42-49), for Atmospheric Chemistry and Physics they should be the main topic of the introduction. The main topic of the introduction is currently Chinese air quality and COVID, but the paper is about the disentangling effects of meteorology from the effects of the COVID lockdown, and so there needs to be more detail on meteorology in China. This is done very well in the introduction to Yin and Zhang (2020); perhaps some more detail specifically on how 2020 meteorology differs from the climatology would distinguish the two studies? You say that variations in the surface wind, boundary layer height and moisture conditions affect air quality, which is not wrong, but specifically what do they typically do in China, when, and where? The literature review also lacks detail; care should be taken to point out explicitly how this paper differs from the large number of other works on the topic. I appreciate this is difficult because of the very large number of very recent publications, but it is definitely possible to do more here.

2. Data description

What technology is usually used to meaure PM2.5 for this dataset? When I tried the URL it didn't work. Please reference the dataset more thoroughly.

3. Model description

This section needs a description of how the model represents aerosol microphysics. The model evaluation presented at the end of this section deserves considerably more detailed study in its own section – what are the biases in the model and how might they affect the subsequent analysis? Unless you can reference other studies evaluating an

identical model configuration?

4. Method to quantify influence of quarantine

Running GEOS-chem for two different emissions scenarios seems like a good idea, and it's good to see that the changes due to meteorology are consistent between years.

However, did you consider the physical justification for a linear decomposition? If we consider, crudely, the Chinese airshed as a simple chemical reactor in steady state, then the linear decomposition would not be obviously appropriate (though it may be a reasonable approximation) since the steady-state concentration is the product of the emissions and the loss lifetime.

Line 99 (minor comment) – I don't fully understand the "the PM2.5 percentage due to changing meteorology". Do you mean "the change in the percentage of PM2.5 due to changing meteorology" here and later in the paragraph?

Line 107 – "the change in emissions resulted in a linear change in air pollution". I don't think this is the message of the very nice Cai et al paper that you cite here. In fact, it is well established that emissions changes often do not lead to linear changes in air pollution, even though I do accept, from the evidence you present, that this is case in China from around 2013 to 2019. The most obvious reason is the sulfate-nitrate-ammonia thermodynamics discussed by Cai et al. Naively, reducing sulfate emissions should reduce concentrations linearly, but reducing nitrate and/or ammonium emissions may not change concentrations at all, or may result in very large decreases in concentrations, depending on the regime (whether saturated by, or limited by, ammonia, for example). Similarly, reducing primary emissions may lead to more new particle formation, as discussed by others, and more secondary aerosol formation, which would also mean the decrease in number concentration is likely sub-linear. Line 197 of the manuscript points this out explicitly. New particle formation wouldn't directly affect changes in mass concentration, but it could have important indirect effects through the size dependence of aerosol dry and wet deposition rates.

So while decreases in concentration may be linear with emissions in specific cases, and does seem to be true in China, this will not be true in general, and should be clarified. Also linearity in previous years, e.g. from 2013 to 2017, does not imply linearity in subsequent years. The linear extrapolation method used therefore brings with it a large uncertainty which should be studied in detail.

5. Results

Line 146: the description is good but some more introductory detail and referencing would be useful. For example, what is the East Asia deep trough? Please supply reference, e.g. Song et al, J. Climate 2016.

Line 149: This is potentially a useful result, but what is the importance of the hygroscopic growth? Its importance surely depends on whether the PM2.5 measurements are of dry or of hydrated particles. If dry particles are measured, hydration might still be important if it affects deposition rates. So what is the difference in humidity and what difference to the size of typical particles would that lead to?

Can you calculate approximate ventilation rates for the boundary layer in the different meteorological conditions, or otherwise increase the level of quantitative detail in lines 140-150, which are currently very qualitative? Can this be used to back up the conclusions about PM2.5? For example, the regression of PM2.5 against "BLH, wind speed, SAT and humidity" done in Yin and Zhang (2020) looks like a nice technique to understand the relationship of air pollution and meteorology, could you do the same thing here for 2020 data? Or at least provide similar numerical detail for what is the BL height and how it varies in the years studied? Is there a role for sea surface temperature here also?

Line 160-165 can you estimate, with quantitative justification, uncertainty ranges for these numbers?

Line 169 – the impacts of COVID-19 quarantines on air quality was weaker south of

30N. This is an interesting conclusion. Could it be related to meteorological differences? Is this consistent with the later statement that in north China, secondary aerosol concentrations increase when primary aerosols decrease? Is that true in south China?

Line 176 what are the reasons for the regional differences?

6. Conclusions

Line 227-240 It is valuable to point out these shortcomings and qualifications for your study. Can you take this further by estimating uncertainties as I suggest above, and speculate what the effect of the interactions between emissions and meteorology would be?

What are the implications of the study for the practice of atmospheric chemistry and physics, beyond those of Yin and Zhang (2020)? Please spell these out in the conclusion.

Figure 1: what is the significance of the red color on the left side of subfigure a)?

Figure 3: state that these figures show simulated data. What is responsible for the increases on the far left of Figure 3c?

Figure 4 please label color bars with units

Detailed language editing is needed throughout the paper.

---

## Referee Comment (RC2) · Anonymous Referee #2 · 30 Sep 2020

General comments: This paper attempted to quantify the effect COVID-19 on the evident PM2.5 decline after removing the influences of climate anomalies and expected routine emissions reductions. Combined with GEOS-Chem model experiments, they used both high and low emission scenarios to simulated the percentages of PM2.5 changes due to meteorological conditions which tended to increase PM2.5 in February 2020, particular in North China. And they further extrapolated the PM2.5 change due to expected routine emission reductions to isolate the decline in PM2.5 concentration due to COVID-19 quarantines in the East of China quantitatively. This study presents some interesting results and could help us better understand the response of air quality to the COVID-19. However, I think the author needs to add some more detailed and rigorous exposition to present their results. Before it can be publishable, I would like the authors

to address my following comments. Major comments Line 65-75 This section requires a more detailed description of the model evaluation. At the end of this section, the author just showed the model could capture the change of meteorological conditions, with high similarly between simulated and observed PM2.5 data. But it is essential that the performance of this model could reproduced the observed true value of PM2.5 concentration. Please evaluate against observation. Line 93 The difference of PM2.5 was linearly decomposed into three parts. I think this is a reasonable approximation, but the author should give more explanation on the rationality of such decomposition. Line 98-99 Please give a detailed calculation method of calculating the percentages of PM2.5 changes due to meteorological conditions. Line 110 The author performed linear extrapolation to obtain PMdR in 2020. The reason to use linear extrapolation here is that the emission reduction caused by the policy is linear, or that the PM2.5 decline caused by emission reduction is approximate linear based on the calculated value of PMdR from 2015 to 2019? The calculated extrapolation results in 2020 are compared with others studies in the latter part of the paper, but please analyze the uncertainty of using this method itself. Line 145 The changes of circulation field, humidity and wind under stagnant weather are analyzed here. Could you give more details about the specific changes in the weather conditions under these stagnant days? Such as boundary layer height and wind speed? Line 167-170 The results of PMdC showed great differences in the north and south regions. What do you think is the cause of this regional difference? Can you give some explanation? Specific comments Line 98 Please explain "the ratio of PMdM of each year/PMdOBS in 2017" more clearly. Are you sure this is divided by "PMdOBS in 2017" here? Or by observed PM2.5 in 2017? Line 101 Keep the same one decimal place. Line 103 Please specify which value is multiplied by this percentage. Line 112 The citation format of this reference is incorrect. Line113 I think it makes more reasonable to write the abbreviation for Beijing-Tian-Hebei here instead of on line 132. Line 124 The abbreviations for North China here and line 122 are repeated. Line 195 Please write NOx here and line 68 in the same way. Figure 1a Clarify what the red and blue bars mean so that the reader can understand this
* * *
Interactive
comment

information. Figure 2 Please give the latitude and longitude range of NC, YRD and HB in the figure caption. Figure 3 The "due to" after each subheading is repeated, leaving out the last three. Figure 4 Add the units of climate elements in the caption (c) and (d). Figure 5 The y-coordinate name is inconsistent with the figure caption. Figure 6 Add the y-coordinate variable name and unit, just like Figure 5. Figure 7a Change the subtitle "PMd" to "PMdOBS" to maintain consistency of expression.

---

## Author Comment (AC1) · 27 Nov 2020

**Reply to Reviewer #1:**

General comments: The authors simulate the decline in $PM_{2.5}$ concentration that resulted from emissions reductions during the COVID-19 pandemic using GEOS-chem. They use 1985 and 2010 emissions to simulate the 2015-20 period. They obtain reasonably good correlations between simulated and observed daily mean $PM_{2.5}$ and show that COVID-19 led to a significant decline. The study is interesting, in the sense that knowing how much $PM_{2.5}$ declined due to COVID-19 after other factors are accounted for is useful, and well-timed. The physical and chemical processes responsible for $PM_{2.5}$ concentrations during COVID are discussed to some extent. In response to my comments during the access review, the authors added two new subfigures elucidating the role of meteorology in generating $PM_{2.5}$, and they added a literature review of chemical mechanisms for the formation of the remaining pollution. **These additions are valuable, but in my opinion further major revisions are still needed before the paper can be published**, as follows:

**1. Abstract and introduction.**
**The abstract and introduction should be refocused towards atmospheric processes. While atmospheric processes are discussed (lines 30-37 and 42-49), for Atmospheric Chemistry and Physics they should be the main topic of the introduction. The main topic of the introduction is currently Chinese air quality and COVID, but the paper is about the disentangling effects of meteorology from the effects of the COVID lockdown, and so there needs to be more detail on meteorology in China. This is done very well in the introduction to Yin and Zhang (2020); perhaps some more detail specifically on how 2020 meteorology differs from the climatology would distinguish the two studies? You say that variations in the surface wind, boundary layer height and moisture conditions affect air quality, which is not wrong, but specifically what do they typically do in China, when, and where? The literature review also lacks detail; care should be taken to point out explicitly how this paper differs from the large number of other works on the topic. I appreciate this is difficult because of the very large number of very recent publications, but it is definitely possible to do more here.**

*Reply:*

Appreciate for your detailed and valuable suggestions, which helped us to improve the main thread of this manuscript.

(1) The main differences between our submission and other publications (with

topic about the impacts of COVID-19 on $PM_{2.5}$) are **whether disentangled effects of meteorology**. Adopting your suggestions, we enhanced related presentations in the Abstract and Introduction.

For example, in the introduction, we added a **detailed analysis of meteorological conditions about typical haze pollution events in the Beijing-Tianjin-Hebei region in December 2016**, and explained how the variations of surface wind, boundary layer height and moisture conditions influenced these severe haze events.

(2) **More specific analysis about the changes in meteorological conditions in February 2020** has also been added. Furthermore, their relationships and regressions against $PM_{2.5}$ were also discussed in lines 175-186, which were also **closely connected with comment 5.3**.

*Revision:*

**Lines 12-14:** Relative to both of February 2017 and climate mean, anomalous southerlies and moister air occurred in the east of China in February 2020, which caused considerable $PM_{2.5}$ anomalies. Thus, it is a must to disentangle the contributions of stable meteorology from the effects of the COVID-19 lockdown.

**Lines 41-44:** Climate variability notably influences the formation and intensity of haze pollution in China……During December 16th-21st 2016, although most aggressive control measures for anthropogenic emissions were implemented, severe haze pollution with $PM_{2.5}$ concentrations $\approx 1100\mu g\ m^{-3}$ still occurred and covered $710,000km^2$. The continuous low surface wind speed of less than $2ms^{-1}$, high humidity above 80% and strong temperature inversion lasting for 132h caused the rebound of wintertime $PM_{2.5}$ in 2016 (Yin and Wang, 2017).

**Lines 48-52:** From February 8 to 13 2020, North China suffered severe pollutions, with maximum daily $PM_{2.5}$ exceeding $200\mu g\ m^{-3}$. During this period, weak southerly surface winds lasted for nearly 5 days, relative humidity was close to 100%, and atmospheric inversion reached more than 10°C. Although pollution emissions from basic social activities have been reduced, heavy pollution still occurred when adverse meteorological conditions characterized by stable air masses appeared (Wang et al.,

2020).

**2. Data description**
**What technology is usually used to measure PM2.5 for this dataset? When I tried the URL it didn't work. Please reference the dataset more thoroughly.**

*Reply:*

The old URL is past-due, and we have updated the **new URL as https://quotsoft.net/air/**. We give a more detailed introduction to the cited dataset and explain the **measurement technology** of $PM_{2.5}$ in this dataset. The $PM_{2.5}$ data were monitored every 5 min using two methods: a tapered element oscillating microbalance (TEOM) and β-rays which were operated **under the China National Quality Control** (HJ/T 193-2005) and (GB3095-2012).

HJ/T 193-2005: Automated methods for ambient air quality monitoring

GB3095-2012: Ambient air quality standards

*Revision:*

**Lines 70-73:** $PM_{2.5}$ concentration data from 2015 to 2020 were acquired from the China National Environmental Monitoring Centre (https://quotsoft.net/air/). The monitoring network expanded from 1500 sites in 2015 to 1640 sites in 2020, covering approximately 370 cities nationwide. The $PM_{2.5}$ data were monitored every 5 min using two methods: a tapered element oscillating microbalance and β-rays which were operated under the China National Quality Control.

**3.Model description**

**This section needs a description of how the model represents aerosol microphysics. The model evaluation presented at the end of this section deserves considerably more detailed study in its own section what are the biases in the model and how might they affect the subsequent analysis? Unless you can reference other studies evaluating an identical model configuration?**

*Reply:*

The description of how the model represents **aerosol microphysics were illustrated in lines 80-85, according to the official website of GEOS-Chem**. The model configurations were **default** and **similar with many previous studies** and the

evaluations of model performances were considerably improved in the following two ways and were documented in a separated paragraph (i.e., Lines 86-101).

(1) With the configuration we used, **comparisons between the observed and simulated** PM$_{2.5}$ concentrations in Feb 2017 were added as new Figure S1a and associated analysis were in lines 89-96. Obviously, mean values of simulated PM$_{2.5}$ were **consistent with the observations** (Figure S1a). The percentage of standard error / mean equals **5.8% (4.6/79.6) in NC, 7.0% (3.9/55.6) in YRD and 5.4% (3.7/70.8) in HB**, indicating the good performance of reproducing the polluted conditions. The biases possibly affected the subsequent results and brought uncertainties to some extent. We also admitted the simulated biases were larger in the south of China, which was consistent with other studies and might **explained the little positive values** in Figure 3c (closely connected with comment 7.2).

[Figure]

**Figure S1a.** Spatial distribution of observed (dots) and GEOS-Chem simulated (shading) PM$_{2.5}$ in February 2017.

Furthermore, the **simulated spatial distribution** was also similar to that of observations in Feb 2017 with **spatial correlation coefficient = 0.78**. The ability of GEOS-Chem to reproduce the daily variations of PM$_{2.5}$ in Feb 2020 was also introduced in the old version as below.

83  changes in February 2020 under a substantial reduction in emissions because of COVID-19 quarantines. In North China (NC),

84  Yangtze River Delta (YRD) and Hubei Province (HB), the correlation coefficients between daily PM$_{2.5}$ observations and

85  simulated data under 2010 (1985) emission scenario reached 0.83 (0.82), 0.67 (0.63), and 0.79 (0.73), respectively. For example,

86  in NC, the simulation could well simulate severe haze events (e.g., from 8–14 and 18–22 February) and good air quality events

87  (e.g., from 15–19 February), reflecting that it has ability to accurately capture the change of meteorological conditions (Fig.

(2) The default configuration of GEOS-Chem were adopted by many previous publications and we also introduced related evaluations in the revised manuscript. Dang and Liao directly evaluated the capacity of models in PM$_{2.5}$ simulations by calculating the normalized mean bias. The simulated spatial patterns of 2013-2017 winter PM$_{2.5}$ were agreed well with the measurements, which was **similar to our evaluations in Figure S1a.** The scatterplot of simulated versus observed **seasonal mean PM$_{2.5}$ concentrations** showed **overestimated** PM$_{2.5}$ concentrations with a normalized mean bias (NMB) of +8.8 % for all grids and an NMB of +4.3 % for BTH (Figure R1a). They also compared the simulated and observed **daily mean PM$_{2.5}$ concentrations** at the Beijing, Shanghai, and Chengdu grids, which represent the three most polluted regions of BTH, YRD, and the Sichuan Basin, respectively. The model has **a low bias in Beijing** with an NMB of **−9.2 %** and is unable to predict the maximum PM$_{2.5}$ concentration in some cases. For Shanghai and Chengdu, the model **has high biases with NMBs of 18.6 % and 28.7 %**, respectively (Figure R1b). This evaluation also showed a bigger simulated bias in the south of China. The model, however, can capture the spatial distributions and seasonal variations of each aerosol species despite of the biases in simulated concentrations.

[Figure]

Figure 2. (a) Spatial distributions of simulated (CTRL, shades) and observed (CNEMC, dots) seasonal (DJF) mean surface-layer PM$_{2.5}$ concentrations (µg m$^{-3}$) averaged over five winters (2013–2017). (b) Scatterplot of simulated versus observed DJF mean PM$_{2.5}$ concentration (µg m$^{-3}$) averaged from 2013 to 2017 for 161 grids in (a), where the green grids are 18 grids located in the BTH (Beijing-Tianjin-Hebei) region. Also shown in (b) are the y = x line (dashed), linear fit (solid line and equation), and values of r and NMB. Here, r is the correlation coefficient between simulated and observed PM$_{2.5}$ concentrations. NMB(normalized mean bias) = ($\sum_{i=1}^{N} (M_i - O_i) / \sum_{i=1}^{N} (O_i)) \times 100\%$, where $O_i$ and $M_i$ are the observed and simulated PM$_{2.5}$ concentrations, respectively. i refers to the ith site, and N is the total number of sites.

Figure 3. Simulated (purple solid line, from CTRL simulation) and observed (green solid line, from the US embassy and CNEMC) daily mean surface-layer PM$_{2.5}$ concentrations (µg m$^{-3}$) for grids of (a) Beijing, (b) Shanghai, and (c) Chengdu over the period when observations are available. Also shown are the threshold of PM$_{2.5}$ = 150 µg m$^{-3}$ (red dashed line); the correlation coefficients (R), and NMB values between observed and simulated daily mean PM$_{2.5}$ concentrations.

**Figure R1**. Key Figures in ***Dang and Liao (2019).***

***Related references:***

Dang, R., and Liao, H.: Severe winter haze days in the Beijing-Tianjin-Hebei region

from 1985 to 2017 and the roles of anthropogenic emissions and meteorology, Atmos. Chem. Phys., 19, 10801–10816, 2019.

***Revision:***

**Line 74:** 2.2 GEOS-Chem description, evaluation and experimental design

**Lines 80-85:** Aerosol thermodynamic equilibrium is computed by the ISORROPIA package, which calculates the gas–aerosol partitioning of the sulfate–nitrate–ammonium system (Fountoukis and Nenes, 2007). Heterogeneous reactions of aerosols include the uptake of $HO_2$ by aerosols (Thornton et al., 2008), irreversible absorption of $NO_2$ and $NO_3$ on wet aerosols (Jacob, 2000), and hydrolysis of $N_2O_5$ (Evans and Jacob, 2005). Two alternate simulations of aerosol microphysics are implemented in GEOS-Chem: the TOMAS simulation (Kodros and Pierce, 2017) and the APM simulation (Yu and Luo, 2009).

**Lines 86-96:** GEOS-Chem model has been widely used to examine the historical changes in air quality in China and quantitatively separate the impacts of physical-chemical processes. Here, we simulated the $PM_{2.5}$ concentrations in February 2017 and evaluated the performance of GEOS-Chem (Figure S1a). The values of mean square error / mean equals were 5.8%, 7.0% and 5.4% in North China (NC), Yangtze River Delta (YRD) and Hubei Province (HB), respectively, indicating the good performance of reproducing the haze-polluted conditions. The absolute biases were larger in the south of China, which was consistent with Dang and Liao (2019). They also compared the simulated and observed daily mean $PM_{2.5}$ concentrations at the Beijing, Shanghai, and Chengdu grids, which had a low bias in Beijing and high biases in Shanghai and Chengdu, respectively. The simulated biases possibly affected the subsequent results and brought uncertainties to some extent. The simulated spatial distribution of $PM_{2.5}$ was also similar to that of observations with spatial correlation coefficient = 0.78. We further verified whether the simulations could capture the roles of meteorological changes in February 2020 under a substantial reduction in emissions because of COVID-19 quarantines…….

**4.Method to quantify influence of quarantine**
**4.1 Running GEOS-chem for two different emissions scenarios seems like a good idea, and it's good to see that the changes due to meteorology are consistent between years. However, did you consider the physical justification for a linear decomposition? If we consider, crudely, the Chinese airshed as a simple chemical reactor in steady state, then the linear decomposition would not be obviously appropriate (though it may be a reasonable approximation) since the steady-state concentration is the product of the emissions and the loss lifetime.**

*Reply:*

The linear decomposition is definitely a **reasonable and feasible approximation** and must have differences with the reality due to complex atmospheric chemical processes (also involving meteorology-emission interactions). The reasons for selecting the linear hypothesis were as follows.

(1) **From 2013 to 2019**, the impacts of emission reduction were approximatively linear, which might related to the **enhanced and reinforced control measures in China**. Because the signal of emissions reduction in China had been **particularly strong since 2013**, it could be easily detected and the assumption of a linear reduction in pollution caused by emission reduction was **applicable in China in the past few years**. This linear approximation was employed by many previous studies (Geng et al. 2017; Zheng et al. 2018) and even by **national assessments aimed to evaluate the effects of *Action Plan of Air Pollution Prevention and Control* from 2013 to 2017** (Geng et al. 2020; Wang et al. 2020). We have introduced the evaluated results in lines 137-142.

(2) After disentangling the effects of meteorology, the variations in $PM_{2.5}$ concentrations also **showed linear change** (Figure 5 in our manuscript), which laterally verified the rationality of linear approximation.

(3) Because of the significantly linear reduction of $PM_{2.5}$ due to changing emissions, the linear decomposition or approximation became reasonable ***in China in recent years*** to some extent.

Certainly, related presentations are lack of physical explanations. We have checked many publications, and all of them have this common problem. We also cannot show you a clear physical justification and only speculated that the obvious linear change due

to emission reductions might be that the control measures in China were particularly enhanced and reinforced. In the revised versions, **we illustrated the linear decompositions were an estimated approach** and must brought some uncertainties due to ignoring the meteorology-emission interactions, the product of emissions and their loss lifetime (Lines 263-267).

***Related references:***

Geng, G., Zhang, Q., Tong, D., Li, M., Zheng, Y., Wang, S., and He, K.: Chemical composition of ambient $PM_{2.5}$ over China and relationship to precursor emissions during 2005–2012, Atmos. Chem. Phys., 17, 9187–9203, https://doi.org/10.5194/acp-17-9187-2017, 2017.

Geng, G., Xiao, Q., Zheng, Y., Tong, D., Zhang, Y., Zhang, X., Zhang, Q., He, H., and Liu, Y.: Impact of China's Air Pollution Prevention and Control Action Plan on PM2.5 chemical composition over eastern China, Sci. China Ser. D., 62, 1872–1884, https://doi.org/10.1007/s11430-018-9353-x, 2020.

Wang, P., Chen, K., Zhu, S., Wang, P., and Zhang, H.: Severe air pollution events not avoided by reduced anthropogenic activities during COVID-19 outbreak, Resour. Conserv. Recy., 158, http://doi:10.1016/j.resconrec.2020.104814, 2020.

Zheng, B., Tong, D., Li, M., Liu, F., Hong, C., Geng, G., Li, H., Li, X., Peng, L., Qi, J., Yan, L., Zhang, Y., Zhao, H., Zheng, Y., He, K., and Zhang, Q.: Trends in China's anthropogenic emissions since 2010 as the consequence of clean air actions, Atmos. Chem. Phys., 18, 14095-14111, 2018

***Revision:***

**Lines 110-112:** As mentioned above, we aimed to examine the impact of the COVID-19 quarantines on $PM_{2.5}$ over the February 2017 level basing on an observational-numerical hybrid method. The observed $PM_{2.5}$ difference in February 2020 ($PMd_{OBS}$) was linearly decomposed into three parts: the impacts of changing meteorology ($PMd_{M}$), expected routine emissions reductions ($PMd_{R}$) and COVID-19 quarantines ($PMd_{C}$), which was a reasonable approximation……

**Lines 263-267:** Furthermore, during the calculation process, the observed $PM_{2.5}$

difference in February 2020 was linearly decomposed into three parts. Although this linear decomposition was reasonable in china in the past few years, we must note that this approximation was lack of considering the meteorology-emission interactions, the product of the emission, the loss lifetime and particularly the sulfate-nitrate-ammonia thermodynamics (Cai et al., 2017), which brought some uncertainties.

**4.2 Line 99 (minor comment) – I don't fully understand the "the PM2.5 percentage due to changing meteorology". Do you mean "the change in the percentage of PM2.5 due to changing meteorology" here and later in the paragraph?**

*Reply:*

What we mean here is that **the percentage of changed PM$_{2.5}$ due to the differences in meteorology** is constant regardless of the emission level. This percentage is the **difference of simulated PM$_{2.5}$ between each year and 2017 under the same emission scenario divided by the simulated PM$_{2.5}$ in 2017**. We have changed the expression to be clearer.

*Revision:*

**Line 119:** Depending on the GEOS-Chem simulations, we found that the percentage of changed PM$_{2.5}$ due to the differences in meteorology remained nearly constant regardless of the emission level (Fig. S2) ……

**4.3 Line 107 – "the change in emissions resulted in a linear change in air pollution". I don't think this is the message of the very nice Cai et al paper that you cite here. In fact, it is well established that emissions changes often do not lead to linear changes in air pollution, even though I do accept, from the evidence you present, that this is case in China from around 2013 to 2019. The most obvious reason is the sulfate-nitrate-ammonia thermodynamics discussed by Cai et al. Naively, reducing sulfate emissions should reduce concentrations linearly, but reducing nitrate and/or ammonium emissions may not change concentrations at all, or may result in very large decreases in concentrations, depending on the regime (whether saturated by, or limited by, ammonia, for example). Similarly, reducing primary emissions may lead to more new particle formation, as discussed by others, and more secondary aerosol formation, which would also mean the decrease in number concentration is likely sub-linear. Line197of the manuscript points this out explicitly. New particle formation wouldn't directly affect changes in mass concentration, but it could have important indirect effects through the size dependence of aerosol dry and wet deposition rates. So while decreases in**

**concentration may be linear with emissions in specific cases, and does seem to be true in China, this will not be true in general, and should be clarified. Also linearity in previous years, e.g. from 2013 to 2017, does not imply linearity in subsequent years. The linear extrapolation method used therefore brings with it a large uncertainty which should be studied in detail.**

*Reply:*

Sorry for the inappropriate citation. Cai et al. paper did not show that emission reduction would lead to linear reduction of air pollution. Just as you said, **from 2013 to 2019, the impacts of emission reduction in China were approximatively linear**. This linear approximation was **employed even by national assessments** aimed to evaluate the effects of *Action Plan of Air Pollution Prevention and Control* from 2013 to 2017 (Geng et al. 2020; Wang et al. 2020).

(1) Due to the implementation of clean air action, control measures have been enhanced and reinforced in China, showing a strong emission reduction signal. Therefore, **the pollutant reduction caused by emission reduction in China from 2013 to 2019 was linear**, which might be **related to the huge emission reduction**. But we didn't check for other areas, maybe not linear reduction. The link has a lot to do with the intensity of emissions reduction. Because the signal of emissions reduction in China had been particularly strong since 2013, it could **be easily detected and showed a linear reduction**.

(2) The effect of emission reduction in February 2020 was calculated as the change of $PM_{2.5}$ caused by **expected** routine emission reduction, **which did not actually happen**, but merely **gave an assessment of the change of $PM_{2.5}$ caused by emission reduction in the case of "if no COVID-19".** Under this hypothetical assessment, the linear change was still tenable.

(3) Furthermore, what we emphasize more was **the effect of total emission reduction ($PMd_R + PMd_C$)**, that was, the common utility of expected routine emissions reductions and COVID-19 quarantines. This quantity was obtained after excluding the effect of meteorological conditions, **which was completely unaffected by linear extrapolation of emission reduction.**

(4) The information revealed by Cai et al. (2017) was valuable and we discussed

**the possible impacts of sulfate-nitrate-ammonia thermodynamics** on our approach in line 267.

*Revision:*

**Lines 130-137:** According to many previous studies, the change in emissions resulted in a linear change in air pollution in China from 2013-2019 (Wang et al., 2020; Geng et al., 2020) which might be related to the huge emission reduction due to the implementation of clean air action. Because the signal of emissions reduction in China had been particularly strong since 2013, it could be easily detected and the assumption of a linear reduction in pollution caused by emission reduction was applicable in China in the past few years. Based on this approximation, we used the method of extrapolation to speculate the impact of routine emission reduction on $PM_{2.5}$. We performed linear extrapolation based on known $PMd_R$ values from 2015 to 2019 to obtain $PMd_R$ in 2020 (STEP 2, Fig. S3). This $PMd_R$ in 2020 was calculated as the change of $PM_{2.5}$ caused by expected routine emission reduction, which did not actually happen, but merely gave an assessment in the case of "if no COVID-19".

**Lines 265-267:** Although this linear decomposition was reasonable in china in the past few years, we must note that this approximation was lack of considering the meteorology-emission interactions, the product of the emission, the loss lifetime and particularly the sulfate-nitrate-ammonia thermodynamics (Cai et al., 2017), which brought some uncertainties.

**5.Results**
**5.1 Line 146: the description is good but some more introductory detail and referencing would be useful. For example, what is the East Asia deep trough? Please supply reference, e.g. Song et al, J. Climate 2016.**

*Reply:*

We have **added the description of the East Asia deep trough** and relevant references.

*Revision:*

**Line 170:** ……the East Asia deep trough, one of the most significant time-mean zonally

asymmetric circulation features in the wintertime Northern Hemisphere (Song et al., 2016), shifted eastwards and northwards than climate mean……

**5.2 Line 149: This is potentially a useful result, but what is the importance of the hygroscopic growth? Its importance surely depends on whether the PM2.5 measurements are of dry or of hydrated particles. If dry particles are measured, hydration might still be important if it affects deposition rates. So what is the difference in humidity and what difference to the size of typical particles would that lead to?**

*Reply:*

Fine aerosols, such as $PM_{2.5}$ particles, will be hygroscopic growth under the environment where the relative humidity is more than 60%, so the measured value without the monitoring instrument to control the relative humidity will be virtual high. When the air is relatively dry, gaseous precursor pollutants could not obviously affect visibility. But in the presence of water molecules, polyphase chemical reactions occurs, and gaseous precursors are oxidized in water droplets or in water carried by particulate matters, accelerating the formation of particulate matter. The conversion rate of $SO_2$ and $NO_2$ into sulfate, nitrate and other particles **increases exponentially with the increase of relative humidity**. Therefore, higher humidity provides a favorable environment for the hygroscopic growth of aerosol particles, which is conducive to the formation of haze pollution and decreasing of visibility.

**What we simply mean to say is the hygroscopic growth of aerosol particles highly reduced the visibility and enhanced the intensity of haze pollution**, rather than impacting the concentration of $PM_{2.5}$. In the revised version, we corrected the sentence to avoid confusions.

*Revision:*

**Lines 173-175:** Physically, the weakening surface winds and strong thermal inversion corresponded to weaker dispersion conditions, and the higher humidity indicated a favorable environment for the hygroscopic growth of aerosol particles to evidently decrease the visibility.

**5.3 Can you calculate approximate ventilation rates for the boundary layer in the different meteorological conditions, or otherwise increase the level of quantitative detail in lines 140-150, which are currently very qualitative? Can this be used to back up the conclusions about PM2.5? For example, the regression of PM2.5 against "BLH, wind speed, SAT and humidity" done in Yin and Zhang (2020) looks like a nice technique to understand the relationship of air pollution and meteorology, could you do the same thing here for 2020 data? Or at least provide similar numerical detail for what is the BL height and how it varies in the years studied? Is there a role for sea surface temperature here also?**

*Reply:*

Thank you for this nice comment. Following it, we not only show more quantitative results, but also **statistically (with observations and regressions) verified** the percentage of changed $PM_{2.5}$ due to the difference in meteorology between 2017 and 2020. We have **added more quantitative analysis** in the revised presentations.

(1) In February 2020, the correlation coefficients of daily $PM_{2.5}$ and BLH, relative humidity, wind speed and SAT in North China were **-0.63, 0.44, -0.45 and 0.46** respectively, all of which **passed the 95% significance test**. Compared with the climate mean status (February 2017), in February 2020 BLH **decreased by 19.5m (34.5m)**, relative humidity **increased by 5% (10.6%)**, and SAT **rose by 1.6°C (0.9°C)** after detrending, which are conductive to the increase of $PM_{2.5}$ concentration.

(2) We used the meteorological data of boundary layer height, relative humidity, surface temperature and wind speed in February 2017 to establish a multiple linear regression equation to fit $PM_{2.5}$. The correlation coefficients between the fitting results and the actual $PM_{2.5}$ concentration in North China, Yangtze River Delta and Hubei reached 0.84, 0.64 and 0.65, all of which **passed the 99% significance test**. Then, we put the **observed meteorological data in February 2020** into the established multiple regression equation to get the predicted $PM_{2.5}$ concentration. Using the regress-predicted value, the percentage of changed $PM_{2.5}$ due to the difference between in meteorology between 2017 and 2020 were re calculated and is 20.7%, -3.2% and 9.5% in NC, YRD and HB, respectively (the hollow column in Figure S2), **which is consistent with and enhanced the robustness of the results obtained by our previous model simulation**.

[Figure]

**Figure S2.** The percentage of changed PM$_{2.5}$ due to the difference in meteorology between 2020 and 2017 by simulated PM$_{2.5}$ with 2010 (red) and 1985 (blue) emission, and regress-fitted PM$_{2.5}$ (hollow). The GEOS-Chem simulations were driven by meteorological conditions in 2017 and 2020 under fixed emissions in 1985 and 2010. The regress-fitted PM$_{2.5}$ was calculated by putting the observed meteorological data in February 2020 into the multiple regression equation fitting PM$_{2.5}$ established by meteorological data in February 2017.

*Revision:*

**Lines 175-186:** Compared with the climate (February 2017) monthly mean, boundary layer height (BLH) decreased by 19.5m (34.5m), surface relative humidity (rhum) increased by 5% (10.6%) and surface air temperature (SAT) rose by 1.6°C (0.9°C) after detrending, which were conductive to the increase of PM$_{2.5}$ concentration in February 2020. Furthermore, the correlation coefficients of daily PM$_{2.5}$ and BLH, rhum, wind speed and SAT in North China were -0.63, 0.44, -0.45 and 0.46, respectively, all of which passed the 95% significance test and indicated importance of meteorology. We used the meteorological data in February 2017 to establish a multiple linear regression equation to fit PM$_{2.5}$. The correlation coefficients between the fitting results and the observed PM$_{2.5}$ concentration in NC, YRD and HB reached 0.84, 0.64 and 0.65, exceeding the 99% significance test. Then, we put the observed meteorological data in February 2020 into this established multiple regression equation to get the predicted

PM$_{2.5}$ concentration. Using the regress-predicted value, the percentage of changed PM$_{2.5}$ due to the differences in Meteorology between 2017 and 2020 were re-calculated and is 20.7%, -3.2% and 9.5% in NC, YRD and HB, respectively (Figure S2), which is consistent with and enhanced the robustness of the results obtained by our previous model simulation.

**5.4 Line 160-165 can you estimate, with quantitative justification, uncertainty ranges for these numbers?**

*Reply:*

We analyzed and discussed the **source of uncertainties**, and also **give the range of bias of GEOS-Chem model simulation**, but the specific range of final uncertainties of is difficult to estimate. Instead, **we can take a step back to give a more comprehensive source of uncertainty in the discussion section** (Lines 258-274).

(1) There is **a certain bias in the simulation by GEOS-Chem model**, and the biases also showed regional differences, which requires further numerical experiments when the emission inventory is updated.

(2) During the calculation process, the observed PM$_{2.5}$ difference in February 2020 was linearly decomposed into three parts. Although this **linear decomposition was reasonable in China in the past few years**, but this approximation was lack of considering the meteorology-emission interactions, the product of the emission, the loss lifetime and particularly the sulfate-nitrate-ammonia thermodynamics (Cai et al., 2017), which brought some uncertainties

(3) The calculation result of the impact of meteorology is **obtained by numerical simulations**, with certain uncertainty. When calculating the expected routine emission reduction in 2020, we use the method of extrapolation. Although the result is consistent with others observational and numerical studies, it is still conjectures rather than true values.

To restrict the possible uncertainties, we **set up some constraints**: 1. The pivotal contribution ratio of changing meteorology were calculated **under two emission levels** and **recalculated by statistical regressed model**; 2. The values of PMd$_M$ and PMd$_R$ were widely **compared to previous studies**.

*Revision:*

**Lines 258-274:** Because of the common update delay of the emission inventory, we employed a combined analysis consisting of observational and numerical methods. We strictly demonstrated the rationality of this method and the results, mainly based on the relatively constant contribution ratio of changing meteorology from GEOS-Chem simulations under the different emissions (Yin and Zhang, 2020). However, there was a certain bias in the simulations by GEOS-Chem model, and the biases also showed regional differences (Dang and Liao, 2019). Therefore, gaps between the assessed results and reality still exist, which requires further numerical experiments when the emission inventory is updated. Furthermore, during the calculation process, the observed $PM_{2.5}$ difference in February 2020 was linearly decomposed into three parts. Although this linear decomposition was reasonable in China in the past few years, we must note that this approximation was lack of considering the meteorology-emission interactions, the product of the emission, the loss lifetime and particularly the sulfate-nitrate-ammonia thermodynamics (Cai et al., 2017), which brought some uncertainties. The actual emission reduction effect is considerable (Fig. 3d), in line with the increasingly strengthened emission reduction policies in recent years. When calculating the $PMd_R$ in 2020, we use the method of extrapolation. Although the result is consistent with others observational and numerical studies (Geng et al., 2019; Zhang et al., 2020; Zhou et al., 2019), it is still conjectures rather than true values. These issues need to be examined in the future studies to unlock respective effects of emissions and meteorological conditions on $PM_{2.5}$ over eastern China. To restrict the possible uncertainties, we set up some constraints: 1. The pivotal contribution ratio of changing meteorology were calculated under two emission levels and recalculated by statistical regressed model; 2. The values of $PMd_M$ and $PMd_R$ were widely compared to previous studies.

**5.5 Line 169 – the impacts of COVID-19 quarantines on air quality was weaker south of 30N. This is an interesting conclusion. Could it be related to meteorological differences? Is this consistent with the later statement that in north China, secondary aerosol concentrations increase when primary aerosols decrease?**

**5.6 Line 176 what are the reasons for the regional differences?**

*Reply:*

The south of 30N is less polluted than the north region, therefore the **background of basic PM$_{2.5}$ concentration is relatively low (Figure S4a)**. In addition, meteorological conditions in the south in February 2020 **had no positive contribution** relative to that in February 2017, which would not lead to the increase of PM$_{2.5}$ concentration. Both of the above two reasons resulted in **a smaller space for PM$_{2.5}$ decrease**. So the PM$_{2.5}$ concentration that can be reduced by COVID-19 in the south is **not as large as** that in North China, and had regional differences.

[Figure]

**Figure S4a.** Observed PM$_{2.5}$ concentrations (unit: μg/m$^3$) in February 2017.

*Revision:*

**Lines 209-212:** Generally, the south region was less polluted than the north, therefore the baseline of PM$_{2.5}$ concentration was relatively lower (Fig. S4a). In addition, meteorological conditions in the south in February 2020 had no positive contribution (Fig. 3a), which would not lead to the increase of PM$_{2.5}$ concentration. These two possible reasons resulted in a smaller space for PM$_{2.5}$ decrease due to COVID-19 quarantines in the south and accompanying regional differences.

**6.Conclusions**

**6.1 Line 227-240 It is valuable to point out these shortcomings and qualifications for your study. Can you take this further by estimating uncertainties as I suggest above, and speculate what the effect of the interactions between emissions and meteorology would be?**

*Reply:*

We can discuss and **make a comprehensive summary of the source of uncertainty** in lines 258-274, but the specific range of uncertainty is difficult to calculate (closely connected with comment 5.4).

About the interaction between emissions and meteorology, it is far away from the topic of this manuscript and we clearly pointed out **this is a new question** in the Section Discussion. Possibly, we solve this question in the near future.

*Revisions:*

**Lines 278-280:** Although the $PM_{2.5}$ dropped much, marked air pollutions also occurred during this unique experiments that the human emissions were sharply closed. This raised new scientific questions, such as changes of atmospheric heterogeneous reactions and oxidability under extreme emission control, quantitative meteorology-emission interactions, and so on.

**6.2 What are the implications of the study for the practice of atmospheric chemistry and physics, beyond those of Yin and Zhang (2020)? Please spell these out in the conclusion.**

*Reply:*

(1) If the COVID-19 epidemic did not occur, the concentrations of $PM_{2.5}$ would **increase up to 1.3–1.7 times** the observations in February 2020. Therefore, the **pollution abatement must continue**. Because of the huge population base in the east of China, the anthropogenic emissions exceeded the atmospheric environmental capacity even during COVID-19 quarantines.

(2) Although the $PM_{2.5}$ dropped much, marked air pollutions **also occurred** during this unique experiments that the human emissions were sharply closed. This raised new scientific questions, such as changes of atmospheric heterogeneous reactions and

oxidability under extreme emission control, quantitative meteorology-emission interactions, and so on. We have added these implications in the Section Conclusion.

*Revision:*

**Lines 275-280:** If the COVID-19 epidemic did not occurred, the concentrations of $PM_{2.5}$ would increase up to 1.3–1.7 times the observations in February 2020 (Figure 6). Therefore, the pollution abatement must continue. Because of the huge population base in the east of China, the anthropogenic emissions exceeded the atmospheric environmental capacity even during COVID-19 quarantines. Although the $PM_{2.5}$ dropped much, marked air pollutions also occurred during this unique experiments that the human emissions were sharply closed. This raised new scientific questions, such as changes of atmospheric heterogeneous reactions and oxidability under extreme emission control, quantitative meteorology-emission interactions, and so on.

**7.1 Figure 1: what is the significance of the red color on the left side of subfigure a)?**

*Reply:*

The **red bars** indicate an **increase** in existing confirmed cases, and the **blue bars** indicate a **decrease**. We make this significance clear in the caption of Figure 1 (a).

*Revision:*

**Line 414:** Figure 1. (a) Variation in existing confirmed cases (bar; red: increase, blue: decrease) and the ratio of accumulated confirmed cases to total confirmed cases (black line) in China…….

**7.2 Figure 3: state that these figures show simulated data. What is responsible for the increases on the far left of Figure 3c?**

*Reply:*

These figures are calculated from **observation data combined with model simulated data**, which mainly depends on the observation data. To avoid confusions, some revisions were included: (1) we have also changed these figures to **be represented as sites**, which are closer to the meaning of the calculation method; (2) In Sec. 2.3, we clearly illustrated the calculations were **based on an observational-**

**numerical hybrid method.**

In the Method and Discussion, we discussed some possible uncertainties. These increases on the far left were **a sort of uncertainties**. These increases were tiny and insignificant, and definitely do **not affected the main results** of our study.

*Revision:*

**Lines 109-110:** As mentioned above, we aimed to examine the impact of the COVID-19 quarantines on $PM_{2.5}$ over the February 2017 level basing on an observational-numerical hybrid method.

**Figure 3.**

[Figure]

**Figure 3.** $PM_{2.5}$ difference (unit: μg/m³) in February between 2020 and 2017 due to (a) changing meteorology ($PMd_M$), (b) expected routine emission reductions ($PMd_R$), (c) the COVID-19 quarantines ($PMd_C$), and (d) due to the total emission reduction ($PMd_E = PMd_R + PMd_C$).

**7.3 Figure 4 please label color bars with units**

*Reply:*

We have added the units to the color bar.

*Revision:*

[Figure]

**Figure 4.** Differences in the observed atmospheric circulation in February between 2020 and 2017, including (a) geopotential potential height at 500 hPa (unit: gpm), (b) wind at 850 hPa (arrows; unit: m/s), surface relative humidity (shading; unit: %). The atmospheric circulations in the stagnant days (e.g., from 8–13 and 19–25 February 2020) were also showed, including (c) geopotential potential height at 500 hPa (shading) and its climate mean in February (contour), and (d) wind at 850 hPa (black arrows), its climate mean (blue arrows) and the increased surface relative humidity (shading, stagnant days minus climate mean).

---

## Author Comment (AC2) · 27 Nov 2020

**Reply to Reviewer #2:**

General comments: This paper attempted to quantify the effect COVID-19 on the evident PM2.5 decline after removing the influences of climate anomalies and expected routine emissions reductions. Combined with GEOS-Chem model experiments, they used both high and low emission scenarios to simulated the percentages of PM2.5 changes due to meteorological conditions which tended to increase PM2.5 in February 2020, particular in North China. And they further extrapolated the PM2.5 change due to expected routine emission reductions to isolate the decline in PM2.5 concentration due to COVID-19 quarantines in the East of China quantitatively. This study presents some interesting results and could help us better understand the response of air quality to the COVID-19. However, I think the author needs to **add some more detailed and rigorous exposition** to present their results. **Before it can be publishable, I would like the authors to address my following comments.**

**Major comments**

Line 65-75 This section requires a more detailed description of the model evaluation. At the end of this section, the author just showed the model could capture the change of meteorological conditions, with high similarly between simulated and observed PM2.5 data. But it is essential that the performance of this model could reproduced the observed true value of PM2.5 concentration. Please evaluate against observation.

**Reply:**

The evaluations of model performances were considerably improved in the following two ways and were documented in a separated paragraph (i.e., Lines 86-101).

(1) With the configuration we used, evaluations between the observed and simulated  $PM_{2.5}$  concentrations in Feb 2017 were added as new Figure S1a and associated analysis were in lines 89-96. Obviously, mean values of simulated  $PM_{2.5}$  were consistent with the observations (Figure S1a). The percentage of standard error / mean equals 5.8% (4.6/79.6) in NC, 7.0% (3.9/55.6) in YRD and 5.4% (3.7/70.8) in HB, indicating the good performance of reproducing the polluted conditions. The absolute biases were larger in the south of China. The simulated spatial distribution was also similar to that of observations in Feb 2017 with spatial correlation coefficient

**Figure S1a.** Spatial distribution of observed (dots) and GEOS-Chem simulated (shading) PM2.5 in February 2017.

Furthermore, the ability of GEOS-Chem to reproduce the daily variations of  $PM_{2.5}$  in Feb 2020 was also introduced in the old version as below.

- changes in February 2020 under a substantial reduction in emissions because of COVID-19 quarantines. In North China (NC),
  Yangtze River Delta (YRD) and Hubei Province (HB), the correlation coefficients between daily PM2.5 observations and
  simulated data under 2010 (1985) emission scenario reached 0.83 (0.82), 0.67 (0.63), and 0.79 (0.73), respectively. For example,
  in NC, the simulation could well simulate severe haze events (e.g., from 8–14 and 18–22 February) and good air quality events
- 87 (e.g., from 15-19 February), reflecting that it has ability to accurately capture the change of meteorological conditions (Fig.

(2) The model configurations were default and similar with many previous studies, which were adopted by many previous publications and we also introduced related evaluations in the revised manuscript. Dang and Liao directly evaluated the capacity of models in PM2.5 simulations by calculating the normalized mean bias. The simulated spatial patterns of 2013-2017 winter PM2.5 were agreed well with the measurements, which was similar to our evaluations in Figure S1a. The scatterplot of simulated versus observed seasonal mean PM2.5 concentrations showed overestimated PM2.5 concentrations with a normalized mean bias (NMB) of +8.8 % for all grids and an NMB of +4.3 % for BTH (Figure R1a). They also compared the simulated and observed daily mean PM2.5 concentrations at the Beijing, Shanghai,

and Chengdu grids, which represent the three most polluted regions of BTH, YRD, and the Sichuan Basin, respectively. The model has **a low bias in Beijing** with an NMB of -9.2 % and is unable to predict the maximum PM2.5 concentration in some cases. For Shanghai and Chengdu, the model **has high biases with NMBs of 18.6** % **and 28.7** %, respectively (Figure R1b). This evaluation also showed a bigger simulated bias in the south of China. The model, however, can capture the spatial distributions and seasonal variations of each aerosol species despite of the biases in simulated concentrations.

Figure R1. Key Figures in Dang and Liao (2019).

**Related references:**

Dang, R., and Liao, H.: Severe winter haze days in the Beijing-Tianjin-Hebei region from 1985 to 2017 and the roles of anthropogenic emissions and meteorology, Atmos. Chem. Phys., 19, 10801–10816, 2019.

**Revision:**

**Lines 86-96:** GEOS-Chem model has been widely used to examine the historical changes in air quality in China and quantitatively separate the impacts of physicalchemical processes. Here, we simulated the PM2.5 concentrations in February 2017 and evaluated the performance of GEOS-Chem (Figure S1a). The values of mean square error / mean equals were 5.8%, 7.0% and 5.4% in North China (NC), Yangtze River Delta (YRD) and Hubei Province (HB), respectively, indicating the good performance of reproducing the haze-polluted conditions. The absolute biases were larger in the south of China, which was consistent with Dang and Liao (2019). They also compared the simulated and observed daily mean PM2.5 concentrations at the Beijing, Shanghai, and Chengdu grids, which had a low bias in Beijing and high biases in Shanghai and Chengdu, respectively. The simulated biases possibly affected the subsequent results and brought uncertainties to some extent. The simulated spatial distribution of  $PM_{2.5}$  was also similar to that of observations with spatial correlation coefficient = 0.78. We further verified whether the simulations could capture the roles of meteorological changes in February 2020 under a substantial reduction in emissions because of COVID-19 quarantines......

**Line 93 The difference of PM2.5 was linearly decomposed into three parts. I think this is a reasonable approximation, but the author should give more explanation on the rationality of such decomposition.**

**Reply:**

The linear decomposition is definitely a **reasonable and feasible approximation** and must have differences with the reality due to complex atmospheric chemical processes. The reasons for selecting the linear hypothesis were as follows.

(1) From 2013 to 2019, the impacts of emission reduction were approximatively linear, which might related to the enhanced and reinforced control measures in China. Because the signal of emissions reduction in China had been particularly strong since 2013, it could be easily detected and the assumption of a linear reduction in pollution caused by emission reduction was applicable in China in the past few years. This linear approximation was employed by many previous studies (Geng et al. 2017; Zheng et al. 2018) and even by national assessments aimed to evaluate the effects of *Action Plan of Air Pollution Prevention and Control* from 2013 to 2017 (Geng et al. 2020; Wang et al. 2020). We have introduced the evaluated results in lines 137-142.

(2) After disentangling the effects of meteorology, the variations in PM2.5 concentrations also **showed linear change** (Figure 5 in our manuscript), which laterally verified the rationality of linear approximation.

(3) Because of the significantly linear reduction of  $PM_{2.5}$  due to changing emissions, the linear decomposition or approximation **became reasonable** *in China in* *recent years* to some extent.

In the revised versions, we illustrated the linear decompositions were an reasonable estimated approach and must brought some uncertainties due to ignoring

the meteorology-emission interactions, the product of emissions and their loss lifetime (Lines 263-267).

**Related references:**

Geng, G., Zhang, Q., Tong, D., Li, M., Zheng, Y., Wang, S., and He, K.: Chemical composition of ambient PM2.5 over China and relationship to precursor emissions during 2005–2012, Atmos. Chem. Phys., 17, 9187–9203, https://doi.org/10.5194/acp-17-9187-2017, 2017.

Geng, G., Xiao, Q., Zheng, Y., Tong, D., Zhang, Y., Zhang, X., Zhang, Q., He, H., and Liu, Y.: Impact of China's Air Pollution Prevention and Control Action Plan on PM2.5 chemical composition over eastern China, Sci. China Ser. D., 62, 1872–1884, https://doi.org/10.1007/s11430-018-9353-x, 2020.

Wang, P., Chen, K., Zhu, S., Wang, P., and Zhang, H.: Severe air pollution events not avoided by reduced anthropogenic activities during COVID-19 outbreak, Resour. Conserv. Recy., 158, http://doi:10.1016/j.resconrec.2020.104814, 2020.

Zheng, B., Tong, D., Li, M., Liu, F., Hong, C., Geng, G., Li, H., Li, X., Peng, L., Qi, J., Yan, L., Zhang, Y., Zhao, H., Zheng, Y., He, K., and Zhang, Q.: Trends in China's anthropogenic emissions since 2010 as the consequence of clean air actions, Atmos. Chem. Phys., 18, 14095-14111, 2018

**Revision:**

**Lines 110-112:** As mentioned above, we aimed to examine the impact of the COVID-19 quarantines on  $PM_{2.5}$  over the February 2017 level basing on an observationalnumerical hybrid method. The observed  $PM_{2.5}$  difference in February 2020 (PMdOBS) was linearly decomposed into three parts: the impacts of changing meteorology (PMdM), expected routine emissions reductions (PMdR) and COVID-19 quarantines (PMdC), which was a reasonable approximation.....

**Lines 263-267:** Furthermore, during the calculation process, the observed  $PM_{2.5}$  difference in February 2020 was linearly decomposed into three parts. Although this linear decomposition was reasonable in china in the past few years, we must note that this approximation was lack of considering the meteorology-emission interactions, the

product of the emission, the loss lifetime and particularly the sulfate-nitrate-ammonia thermodynamics (Cai et al., 2017), which brought some uncertainties.

**Line 98-99 Please give a detailed calculation method of calculating the percentages of PM2.5 changes due to meteorological conditions.**

**Reply:**

We use the **simulated PM2.5 data** driven by changing meteorology with two fixedemissions (1985 and 2010). This percentage is the **difference of simulated PM2.5 between each year and 2017** under the same emission scenario **divided by the simulated PM2.5 in 2017**. We have added this detailed description in the text.

**Revision:**

**Lines 120-121:** This percentage was the difference of simulated  $PM_{2.5}$  between each year and 2017 under the same emission scenario divided by the simulated  $PM_{2.5}$  in 2017.

Line 110 The author performed linear extrapolation to obtain PMdR in 2020. The reason to use linear extrapolation here is that the emission reduction caused by the policy is linear, or that the PM2.5 decline caused by emission reduction is approximate linear based on the calculated value of PMdR from 2015 to 2019? The calculated extrapolation results in 2020 are compared with others studies in the latter part of the paper, but please analyze the uncertainty of using this method itself.

**Reply:**

From 2013 to 2019, the impacts of emission reduction on PM2.5 in China were approximatively linear, which might due to the control measures in China were particularly enhanced and reinforced. This linear approximation was employed even by national assessments aimed to evaluate the effects of *Action Plan of Air Pollution Prevention and Control* from 2013 to 2017 (Geng et al. 2020; Wang et al. 2020).

(1) Due to the implementation of clean air action, control measures have been enhanced and reinforced in China, showing a strong emission reduction signal. Therefore, **the pollutant reduction caused by emission reduction** in China from 2013 to 2019 **was linear**, which might be related to the huge emission reduction. The link has a lot to do with the intensity of emissions reduction. Because the signal of emissions reduction in China had been particularly strong since 2013, it could **be easily detected and showed a linear reduction**. (2) The effect of emission reduction on PM2.5 in February 2020 was calculated as the change of  $PM_{2.5}$  caused by expected routine emission reduction, which did not actually happen, but merely gave an assessment of the change of  $PM_{2.5}$  caused by emission reduction in the case of "if no COVID-19". Under this hypothetical assessment, the linear change was still tenable.

(3) Furthermore, what we emphasize more was the effect of total emission reduction ( $PMd_R + PMd_C$ ), that was, the common utility of expected routine emissions reductions and COVID-19 quarantines. This quantity was obtained after excluding the effect of meteorological conditions, which was completely unaffected by linear extrapolation of emission reduction.

(4) The calculated extrapolation results in 2020 is consistent with others observational and numerical studies, but we must note that it is still **conjectures rather than true values**, which was lack of considering the meteorology-emission interactions and the sulfate-nitrate-ammonia thermodynamics, which brought some uncertainties. We have added **the analyze of this uncertainty** in line 267.

**Revision:**

Lines 130-137: According to many previous studies, the change in emissions resulted in a linear change in air pollution in China from 2013-2019 (Wang et al., 2020; Geng et al., 2020) which might be related to the huge emission reduction due to the implementation of clean air action. Because the signal of emissions reduction in China had been particularly strong since 2013, it could be easily detected and the assumption of a linear reduction in pollution caused by emission reduction was applicable in China in the past few years. Based on this approximation, we used the method of extrapolation to speculate the impact of routine emission reduction on PM2.5. We performed linear extrapolation based on known PMdR values from 2015 to 2019 to obtain PMdR in 2020 (STEP 2, Fig. S3). This PMdR in 2020 was calculated as the change of PM2.5 caused by expected routine emission reduction, which did not actually happen, but merely gave an assessment in the case of "if no COVID-19". Under this hypothetical assessment, the linear change was still tenable. Lines 265-267: .....we must note that this approximation was lack of considering the meteorology-emission interactions, the product of the emission, the loss lifetime and particularly the sulfate-nitrate-ammonia thermodynamics (Cai et al., 2017), which brought some uncertainties.

**Line 145 The changes of circulation field, humidity and wind under stagnant weather are analyzed here. Could you give more details about the specific changes in the weather conditions under these stagnant days? Such as boundary layer height and wind speed?**

**Reply:**

Appreciate for your valuable suggestion. We not only show more **quantitative** results, but also statistically (with observations and regressions) verified the percentage of changed  $PM_{2.5}$  due to the difference in meteorology between 2017 and 2020. We have added more quantitative analysis in the revised presentations.

(1) In February 2020, the correlation coefficients of daily PM2.5 and BLH, relative humidity, wind speed and SAT in North China were -0.63, 0.44, -0.45 and 0.46 respectively, all of which **passed the 95% significance test**. Compared with the climate mean status (February 2017), in February 2020 BLH decreased by 19.5m (34.5m), relative humidity **increased by 5% (10.6%)**, and SAT **rose by 1.6°C (0.9°C)** after detrending, which are conductive to the increase of PM2.5 concentration.

(2) We used the meteorological data of boundary layer height, relative humidity, surface temperature and wind speed in February 2017 to establish a multiple linear regression equation to fit  $PM_{2.5}$ . The correlation coefficients between the fitting results and the actual  $PM_{2.5}$  concentration in North China, Yangtze River Delta and Hubei reached 0.84, 0.64 and 0.65, all of which **passed the 99% significance test**. Then, we put the **observed meteorological data in February 2020** into the established multiple regression equation to get the predicted  $PM_{2.5}$  concentration. Using the regress-predicted value, the percentage of changed  $PM_{2.5}$  due to the difference between in meteorology between 2017 and 2020 were re calculated and is 20.7%, -3.2% and 9.5% in NC, YRD and HB, respectively (the hollow column in Figure S2), which is consistent with and enhanced the robustness of the results obtained by our

previous model simulation.

---

## Author Response (AR2)

**Reply to Reviewer #2:**

Comments to the Author: This paper documents the PM2.5 changes as a result of COVID-19 in China. Following my previous comments the authors have made many useful revisions and the paper is much improved. I recommend it **be accepted** subject to the following mainly **minor comments being addressed**. Overall I think my revisions are minor, but they are important:

**1. Figure S1 is very helpful and should be added to the main text. With reference to Figure S1a, the authors should comment on the substantial underestimate of the PM2.5 in northern China (not just in Beijing) and in the westernmost parts of your figure at 35N, and calculate a normalized mean bias from their figure and add it to the text with a comparison to the number obtained by Dang and Liao (2019).**

**They should also discuss Figure S1b in detail in the text, rather than mentioning it in passing. It shows that on average the 2010 emissions substantially overestimate PM2.5 everywhere and the 1985 emissions still overestimate PM2.5, especially in the south, but agree better with the measurements. This is a good justification for using the 1985 inventory, which the authors should explain at line 119, and then add a substantial new paragraph to the text detailing their evaluation.**

*Reply:*

Appreciate for your detailed and valuable comments. In terms of the evaluation of GEOS-Chem model, we have **made a more complete description based on your comments** and **added Figure S1 to the main text as Figure 2**.

(1) Dang and Liao (2019) compared the simulated and observed daily mean PM$_{2.5}$ concentrations at the Beijing with a normalized mean bias (NMB) of -9.2%. The simulations in February 2017 in this study substantially underestimated the PM$_{2.5}$ in northern China with a normalized mean bias (NMB) of -3.0%. Among them, the NMB in The Beijing-Tianjin-Hebei region was -3.3%. However, in the Fenwei plain (the westernmost parts of the figure at 35N), the underestimation was even more pronounced, with NMB reaching -16.3%.

(2) In North China, Yangtze River Delta and Hubei Province, the correlation coefficients between daily PM$_{2.5}$ observations and simulated data under 2010 (1985) emission scenario reached 0.83 (0.82), 0.67 (0.63), and 0.79 (0.73), respectively. The correlation coefficients under 2010 emission scenario were all higher than that under emission scenario maybe due to the **emissions from each sector in 2010 were more similar to recent years**, which was more reasonable. Therefore, we selected the percentages due to different meteorology between 2020 and 2017 calculated under the 2010 emission scenario, instead of making the selection based on the simulation results of the real $PM_{2.5}$ value, which was also **mentioned in the following text** as below.

$PM_{2.5}$ between each year and 2017 under the same emission scenario divided by the simulated $PM_{2.5}$ in 2017. For example, the percentages due to different meteorology between 2020 and 2017 were 22.1% (21.4%), −1.2% (−0.7%) and 9.0% (8.2%) in

NC, YRD and HB under the low (high) emissions (Fig. S2). The percentage under 2010 emission scenario was selected as the final percentage because the emissions from each sector in 2010 were more similar to recent years, and thus was more reasonable. Then, through multiplying the 2017 observation by this percentage , $PMd_M$ can be quantified in each simulation

The evaluation of the simulated $PM_{2.5}$ concentration under 2010 emission and 1985 emission in February 2020 was also introduced in the old version as below. In the revised version, we present this section **as a new and separate paragraph** and give a more detailed evaluation and explanation.

***Revision:***

**Lines 92-98:** The absolute biases were larger in the south of China, which was consistent with Dang and Liao (2019). They also compared the simulated and observed daily mean $PM_{2.5}$ concentrations at the Beijing, Shanghai, and Chengdu grids, which had a low bias in Beijing with a normalized mean bias (NMB) of -9.2% and high biases with NMBs of 18.6% and 28.7% in Shanghai and Chengdu, respectively. The simulations in February 2017 in this study substantially underestimated the $PM_{2.5}$ in NC with an NMB of -3.0% (Fig. 1a). Among them, the NMB in The Beijing-Tianjin-Hebei region was -3.3%. However, in the Fenwei plain, the underestimation was even more pronounced, with NMB reaching -16.3%.

**Lines 102-109:** In NC, YRD and HB, the correlation coefficients between daily $PM_{2.5}$ observations and simulated data under 2010 (1985) emission scenario reached 0.83 (0.82), 0.67 (0.63), and 0.79 (0.73), respectively (Fig.1b), and could capture the maximum and minimum $PM_{2.5}$ concentrations……. The correlation coefficients under 2010 emission scenario were all higher than that under 1985 emission scenario maybe due to the emissions from each sector in 2010 were more similar to recent years, which was more reasonable.

**2. Rephrase first sentence: "blew China" is not conventional English. "swept through China" would be OK. In general, the quality of the written English could still be improved significantly in several other places not mentioned below, I recommend the authors seek advice from colleagues if at all possible, or assistance from the copy-editors.**

*Reply:*

Thank you for this detailed suggestions. We have rephrased "blew China" to "swept through China" and checked the quality of the written English of the whole text.

*Revisions:*

**Line 23:** The COVID-19 pandemic devastatingly swept through China in the beginning of 2020……

**3. Line 50-52 are you referring to the same rebound. Rephrase to avoid repetition.**

*Reply:*

These were **two different rebound**. One was the severe haze events occurring in 2016 December, indicating a rebound of $PM_{2.5}$ comparing to 2014-2015, and the other was the rebound of $PM_{2.5}$ in winter 2018 comparing to 2017 under the same intensified regional air pollution preventions. We have rephrased the explanation to make it clearer.

*Revision:*

**Lines 43-47:** The continuous low surface wind speed of less than $2ms-1$, high humidity above 80% and strong temperature inversion lasting for 132h caused the serious haze event in 2016 (Yin and Wang, 2017). In winter 2017, the air quality in North China largely improved; however, the stagnant atmosphere in 2018 resulted in a major $PM_{2.5}$ rebound comparing to 2017 by weakening transport dispersion and enhancing the chemical production of secondary aerosols (Yin and Zhang 2020).

**4. Line 92 which aerosol microphysics is used here?**

*Reply:*

According to the official website of GEOS-Chem, in the mechanism we run, these two alternate simulations of aerosol microphysics were both simulated. We have explained in the text.

*Revision:*

**Lines 85-86:** Two alternate simulations of aerosol microphysics are implemented in

GEOS-Chem: the TOMAS simulation (Kodros and Pierce, 2017) and the APM simulation (Yu and Luo, 2009), which were both simulated in the experiments.

**5. Line 103 please be quantitative, add the magnitude of the biases to the text.**

*Reply:*

Dang and Liao (2019) compared the simulated and observed daily mean PM2.5 concentrations at the Beijing, Shanghai, and Chengdu grids, which had a low bias in Beijing with a normalized mean bias (NMB) of **-9.2%** and high biases with NMBs of **18.6%** and **28.7%** in Shanghai and Chengdu, respectively. We have added the specific biases to the text.

*Revisions:*

**Lines 94-96:** They also compared the simulated and observed daily mean $PM_{2.5}$ concentrations at the Beijing, Shanghai, and Chengdu grids, which had a low bias in Beijing with a normalized mean bias (NMB) of -9.2% and high biases with NMBs of 18.6% and 28.7% in Shanghai and Chengdu, respectively.

**6. Line 203 and 206 need to explain how the significance testing was carried out.**

*Reply:*

We use *t* **test** which depends on *t* distribution. According to querying the critical value table of correlation coefficient by reliability and degree of freedom, the critical correlation coefficient that passes the *t* test is obtained. If the calculated correlation coefficient is greater than the critical correlation coefficient, it means passing the significance *t* test of the corresponding reliability. We have explained in the text that we used *t* test to test the significance of correlation coefficients.

*Revision:*

**Line 188:** …… all of which passed the 95% significance test using *t* test method ……

**Line 191:** …… exceeding the 99% significance test using *t* test method ……

**7. Line 262 "break-off transportation" needs rephrasing.**

*Reply:*

We have rephrased the "break-off transportation" to "the disruption of transportations".

*Revision:*

**Line 245:** Because of the disruption of transportations……

**8. Line 296 "approximation was lack of considering"-> "approximation did not consider"**

*Reply:*

We have changed "approximation was lack of considering" to "approximation did not consider".

*Revisions*

**Line 275:** ……we must note that this approximation did not consider the meteorology-emission interactions……

**9. The word 'conjecture' is inappropriate, I would comment that the technique introduces uncertainty.**

*Reply:*

We have changed "conjecture" to "**estimated value**", which could show our meaning appropriately.

*Revision:*

**Line 280:** ……it is still estimated value rather than true value……

**10. Figure 1b caption: specify time period for ratios**

*Reply:*

The time period for ratios is **until the end February**. We have specified it in the caption.

*Revisions:*

[revised manuscript text omitted]